# FAST EQUILIBRIUM OF SGD IN GENERIC SITUATIONS

**Zhiyuan Li** [*]
Toyota Technological Institute at Chicago
zhiyuanli@ttic.edu

**Yi Wang** [*]
Johns Hopkins University
ywang261@jhu.edu

**Zhiren Wang** [*]
Pennsylvania State University
zhirenw@psu.edu

## ABSTRACT

Normalization layers are ubiquitous in deep learning, greatly accelerating optimization. However, they also introduce many unexpected phenomena during training, for example, the Fast Equilibrium conjecture proposed by (Li et al., 2020), which states that the scale-invariant normalized network, when trained by SGD with $\eta$ learning rate and $\lambda$ weight decay, mixes to an equilibrium in $\tilde{O}(\frac{1}{\eta\lambda})$ steps, as opposed to classical $e^{O((\eta\lambda)^{-1})}$ mixing time. Recent works by Wang & Wang (2022); Li et al. (2022c) proved this conjecture under different sets of assumptions. This paper aims to answer the fast equilibrium conjecture in full generality by removing the non-generic assumptions of Wang & Wang (2022); Li et al. (2022c) that the minima are isolated, that the region near minima forms a unique basin, and that the set of minima is an analytic set. Our main technical contribution is to show that with probability close to 1, in exponential time trajectories will not escape the attracting basin containing their initial position.

## 1 INTRODUCTION

Normalization layers are ubiquitous and play a fundamental role in modern deep learning, *e.g.*, Batch Normalization (Ioffe & Szegedy, 2015), Group Normalization (Wu & He, 2018), Layer Normalization (Ba et al., 2016), and Weight Normalization (Salimans & Kingma, 2016). Normalization layers not only greatly facilitate optimization and improve trainability, it also brings intriguing new optimization behaviors to neural networks. For example, Li & Arora (2020) showed that normalized networks can be trained by SGD with exponentially increasing learning rates, because training with exponentially increasing learning rates turns out to be equivalent to training with constant learning rates but with weight decay turned on, as shown in (2). Here $x_k \in \mathbb{R}^d$ is the parameter of a neural network after the $k$-th step and is updated by

$$x_{k+1} \leftarrow (1 - \lambda)x_k - \eta\nabla L_{\mathcal{B}_k}(x_k), \tag{1}$$

where $\lambda$ and $\eta$ are respectively the weight decay parameter and the learning rate, and $L_{B_k}$ is the loss function evaluated using a randomly chosen mini-batch $\mathcal{B}_k$.

The result of Li & Arora (2020) holds not only for normalized networks but more broadly for all scale invariant training losses, which is a popular abstraction of normalized networks in optimization analysis. Mathematically, scale invariance refers to the following property of the loss: $L_{\mathcal{B}}(cx) = L_{\mathcal{B}}(x)$, $\forall c > 0, x \in \mathbb{R}^d$ and every batch $\mathcal{B}$.

Later, Li et al. (2020); Wan et al. (2021) discovered that it is the *intrinsic learning rate* $\eta\lambda$ that controls the long-term convergence behavior for SGD on scale invariant loss with weight decay, (1). The approach that Li et al. (2020) takes to study (1) is to approximate by the stochastic differential equation (SDE) model Li et al. (2017; 2019), which is quite common in literature.

$$\mathrm{d}X_t = (-\eta\nabla L(X_t) - \eta\lambda X_t)\mathrm{d}t - \eta\sigma(X_t)\mathrm{d}B_t^K. \tag{2}$$

---

[*]The authors are listed alphabetically.

Here $L$ is the average $\frac{1}{K}\sum_{k=1}^{K} L_{\mathcal{B}_k}$ over all random batches, $\sigma = \frac{1}{\sqrt{K}}\left(\nabla L_{\mathcal{B}_k} - \nabla L\right)_{k=1}^{K}$ is a $d \times K$ matrix, and $B_t^K$ is the $K$-dimensional Wiener process. Scale invariance of $L_{\mathcal{B}}$ implies that $L$ is scale-invariant and that $\sigma$ is $(-1)$-homogeneous, i.e.

$$L(cx) = L(x), \sigma(cx) = c^{-1}\sigma(x), \forall c > 0, x \in \mathbb{R}^d. \tag{3}$$

Li et al. (2020) further proposed the following Fast Equilibrium Conjecture for the SDE approximation of SGD.

**Conjecture 1.1.** *[Fast Equilibrium Conjecture] (Li et al., 2020) If $F(X, \texttt{input})$ denotes the output of a neural network* NN *with parameters $X$, and $X_t$ denotes the value of SDE (2) at time $t$, starting from initial parameter $X_0$. Suppose* NN *has normalization steps so that the $F(X, \texttt{input})$ is scale-invariant in $X$, i.e. $F(X, \texttt{input}) = F(cX, \texttt{input})$ for all $c > 0$. Then for all input values* input, *the probability distribution of $F(X_t, \texttt{input})$ stabilizes to an equilibrium in $O(\frac{1}{\eta\lambda})$ steps of SGD updates.*

Experiments where the empirically observed rates of convergence are polynomial were contained in the original paper Li et al. (2020) where the Fast Equilibrium Conjecture was first asked. The rate $O(\frac{1}{\eta\lambda})$ is considered to be fast because according to Langevin dynamics, the time it takes to converge to the Gibbs equilibrium is of exponential order $e^{O((\eta\lambda)^{-\frac{1}{2}})}$. This can be done by following a similar analysis to those in (Bovier et al., 2004; Shi et al., 2020). The works by (Bovier et al., 2004) and (Shi et al., 2020) dealt with models without normalization, and the convergence times there are of order $e^{O((\eta\lambda)^{-1})}$. Using the similar method as in (Bovier et al., 2004) and (Shi et al., 2020), when normalization is used the convergence time can be shown to be of order $e^{O((\eta\lambda)^{-\frac{1}{2}})}$. This is because Li et al. (2020) proved that the intrinsic learning rate $\eta\lambda$ is replaced by an effective learning rate ($\gamma_t^{-\frac{1}{2}}$ in Li et al. (2020)) for the renormalized parameter vector, which is of order $O((\eta\lambda)^{\frac{1}{2}})$.

The recent paper (Li et al., 2022c), using a mathematical framework from (Li et al., 2022b), established the fast equilibrium conjecture for $\eta\lambda \to 0$ under a mixed set of generic and non-generic assumptions. See (Damian et al., 2021), (Gu et al., 2022) for more work on analyzing the dynamics of SGD near the manifold of minimizers. The goal of the current paper is to remove the non-generic ones and thus provide a general proof in the aforementioned range of parameters.

### 1.1 NOTATIONS AND ASSUMPTIONS

To introduce assumptions from previous authors as well as our results, we need to set up a few notations first. Let $\Gamma \subseteq \mathbb{R}^d \backslash \{0\}$ be the set of local minima of $L$. Notice that by (3), $\Gamma$ is a cone, i.e. $x \in \Gamma$ if and only if $cx \in \Gamma$ for all $c > 0$. For all $r > 0$, write $\Gamma_r = \{x \in \Gamma : |x| = r\}$. In particular $\Gamma_1$ is a subset of the unit sphere $\mathbb{S}^{d-1} = \{|x| = 1\}$.

In general, $\Gamma$ may have multiple connected components. Decompose $\Gamma = \bigsqcup \Gamma^i$ where each $\Gamma^i$ is a connected cone. We then write $\Gamma_r^i = \Gamma^i \cap \{|x| = r\}$. Then $\Gamma_1^i$ are the connected components of $\Gamma_1$. In particular, there are only finitely many $\Gamma^i$'s and we index them by $i = 1, \cdots, m$.

In addition to the scaling properties (3) guaranteed by the use of normalization, (Li et al., 2022c) made certain assumptions, which we will need in the following.

**Assumption 1.2.** *The functions $L$ and $\sigma$ satisfy:*

**(i). (Scale invariance)** *The scaling rules in (3) hold.*

**(ii). (Regular critical locus)** *Each loss function $L_{\mathcal{B}_k}$ is $C^4$ on $\mathbb{R}^d \backslash \{0\}$, the critical points of $L$ form a $C^2$ submanifold $\Omega$. [1] For all $x \in \Omega$, $\nabla^2 L(x)$ is of rank $d - \dim T_x\Omega$.*

**(iii). (Controllability)** *For all $x \in \Gamma_1$, $\text{span}\{\partial\Phi(x)\sigma_k(x)\}_{k=1}^{K} = T_x\Gamma_1$. Here and below, $\Phi(x) = \lim_{t\to\infty} X_t$, with $X_t$ being the solution to the deterministic gradient descent $X_t = -\nabla L(X_t)$ with initial value $x$.*

By (Arora et al., 2022, Lemma B.15), under Assumption 1.2.(i) & (ii), $\Phi$ is well defined and $C^2$-differentiable on a neighborhood of $\Gamma$ as long as $L$ is $C^4$ differentiable. We also note that in general

---

[1] Different connected components of $\Omega$ are not required to have the same dimension.

the noise structure does affect the convergence rate. But as long as Assumption 1.2 (iii) is satisfied, the noise structure won't affect the asymptotic order of the convergence rate.

On the other hand, we will not need the following assumptions.

**Assumption 1.3.** $\Gamma$ *satisfies:*

**(i). (Unique basin)** $\Gamma_1$ *is compact and connected;*

**(ii). (Analyticity)** $\Gamma$ *is a real analytic manifold and* $\operatorname{Tr}\Sigma$ *is a real analytic function on* $\mathbb{R}^d\backslash\{0\}$ *where* $\Sigma = \sigma\sigma^\top$.

Restricting to an attracting basin $U$ of $\Gamma$ and assuming both Assumptions 1.2 and 1.3, Li et al. (2022c) proved Conjecture 1.1 when $\lambda\eta \to 0$ in the natural range of $\eta \le O(\lambda) \le O(1)$ and the parameter $X_0$ is initialized within $U$. Note that since $U$ is an attracting basin, $\Omega$ and $\Gamma$ coincide in $U$ and thus Assumption 1.2 is equivalent to (Li et al., 2022c, Assumption 2.1) for the purpose of that paper. Note that $\Gamma$ is always a submanifold of $\Omega$.

**Remark 1.4.** *All three assumptions in Assumption 1.2 are very natural for the following reasons:*

• *As remarked earlier, the scale-invariance (3) is a consequence of the use of normalization steps inside neural networks.*

• *It is a widely used assumption, at least in the case of local minimizers, that the locus is a manifold for overparametrized neural networks, for example in (Fehrman et al., 2020; Arora et al., 2022; Li et al., 2022b). For the locus of global minimizers, this assumption was proved by Cooper (2021). As remarked in (Li et al., 2022b; Cooper, 2021), a main reason for the local minimizers to form manifolds is the overparametrization of modern neural networks. Şimşek et al. (2021) further identifies the reason as symmetries arising from overparametrization. In fact, they studied loci of critical points that are not necessarily minima and proved that symmetry-induced critical points form a manifold that satisfies Assumption 1.2.(ii).*

• *The philosophy behind Assumption 1.2.(iii) is that the generation of random batches in training is independent of the aforementioned symmetries in the setup of the neural network, and thus generically should not live in subspaces that are invariant under such symmetries. In particular, the same symmetries are generically capable of move the noises from the random batches to span a tangent space of the same dimension as that of the local manifold of critical points.*

**Remark 1.5.** *On the other hand, both conditions in Assumption 1.3 are non-generic:*

• *Certain evidences from (Draxler et al., 2018) suggests that all local minima appearing in realistic training generically come from connected relatively flat region of small variation in height, so empirically Assumption 1.3.(i) could be a reasonable approximate assumption. However, the experiments in (Draxler et al., 2018, Fig. 5) shows at the same time that in many settings, this region is not completely flat and contains non-trivial saddle points. In particular, there could be multiple disconnected basins. In light of these, it is more reasonable work in the absence of assumption Assumption 1.3.(i).*

• *The analyticity of the $\Gamma$ and $\operatorname{Tr}\Sigma$ depends on that of the activation functions chosen in the neural network. While many popular activation functions are real analytic, one may always choose to use functions that are differentiable but not analytic, in which case Assumption 1.3.(ii) is in general not guaranteed.*

In this paper, we will give a general proof of Conjecture 1.1 in the same natural range as in (Li et al., 2022c), assuming only the generic conditions from Assumption 1.2. In particular, we will introduce two arguments that respectively remove both hypothesis (Unique basin and Analyticity) in Assumption 1.3.

Here we would like to provide comments on why Assumption 1.3 are restrictive. Assuming analyticity is restrictive because the regularity of the loss function is decided by that of the activation function. Even though popular activation functions such as Sigmoid are analytic, a priori one could use smooth but not analytic functions. The one basin assumption is restrictive as we do not see empirical evidence of proof that $L$ only has one basin. In fact, the experiments at the end of the paper suggests that there are multiple basins.

We would also like to give remarks on why the three assumptions in Assumption 1.2 are essential. (i) is essential because without this assumption, the SDE would not be equivalent to a SDE on the sphere $\mathbb{S}^{d-1}$, which is crucial to our analysis. Without this assumption, similar analysis can probably be formulated on $\mathbb{R}^d$ instead of the $\mathbb{S}^{d-1}$ coordinate but there will be new technical obstacles to overcome. Since the original fast equilibrium conjecture was asked for normalized neural nets, we restrict our study to the current setting. (ii) is important because if not a trajectory may stay near a critical point (for example a saddle point) for a very long period of time, it would not be able to converge within a polynomial time. Finally, the reason why we need (iii) is that if the span is not the whole tangent space, but instead a subspace of the tangent space, then the diffusion will be restrained to this subspace, which a priori may be very fractal and existing mathematical theory is not sufficient to guarantee a unique equilibrium in limit.

## 1.2 MULTIPLE EQUILIBRIA WHEN BASIN IS NOT UNIQUE

It is worthy to explain in more details what happens when the basin is not unique, i.e $\Gamma$ has multiple connected components and Assumption 1.3.(i) fails. In this situation, our analysis generalizes the work of Wang & Wang (2022) and reveals a three-stage equilibrium phenomenon. The most important property of this phenomenon is the mismatching between practical training and theoretical bounds: the equilibrium distribution of network parameters observed in the time window under a realistic budget is both local in space and temporary in time. It is concentrated near the bottom of the same attracting basin containing the initial parameter, and differs from the eventual global Gibbs equilibrium that the distribution of parameters will eventually converge to in exponentially long time. This phenomenon interprets the gap between the empirically based Conjecture 1.1 and the previous theoretical estimate from e.g. (Bovier et al., 2004; Shi et al., 2020). See (Frankle et al., 2020), (Gupta et al., 2019) for more work about the iterates stay in the same basin for a significant amount of time when starting from the same initialization.

The major short-come of (Wang & Wang, 2022) is that, while not relying on the uniqueness of the basin, the arguments therein are subject to other non-generic assumptions, namely: (1) all basins are isolated points; (2) the noise $\sigma$ is a standard isotropic Gaussian noise.

Our methods allow to remove these assumption simultaneously together with Assumption 1.3. This is made possible by avoiding using the semi-classical analysis of spectra of differential operators, which was developed by Simon (Simon, 1983) and used in an essential way by previous authors in (Bovier et al., 2004; Shi et al., 2020; Wang & Wang, 2022). Instead, our method is purely probabilistic and predicts that the exiting time from a given basin is exponentially long. This method is based on an adaptation of the large deviation principle of Dembo & Zeitouni (2010).

## 2 STATEMENT OF MAIN RESULTS

### 2.1 PRELIMINARIES ON SDE MODEL

A polar coordinate system has been adopted in (Li et al., 2020) to study the SDE model (2). For this purpose, denote by $\overline{X}_t = \frac{X_t}{|X_t|}$ the unit renormalization of $X_t$, and $\gamma_t = |X_t|^4 \eta^{-2}$. By (Li et al., 2020, Theorem 5.1), (2) is equivalent to

$$\mathrm{d}\overline{X}_t = -\gamma_t^{-\frac{1}{2}}\Big(\nabla L(\overline{X}_t)\mathrm{d}t + \sigma(\overline{X}_t)\mathrm{d}B_t^K\Big) - \frac{1}{2}\gamma_t^{-1}\operatorname{Tr}\Sigma(\overline{X}_t)\overline{X}_t\mathrm{d}t; \tag{4}$$

$$\frac{\mathrm{d}\gamma_t}{\mathrm{d}t} = -4\eta\lambda\gamma_t + 2\operatorname{Tr}\Sigma(\overline{X}_t). \tag{5}$$

Recall that $\Sigma = \sigma\sigma^\top$ is a $d \times d$ positive semidefinite symmetric matrix.

One may view the motion of $\overline{X}_t$ as an intrinsic one inside the unit sphere $\mathbb{S}^{d-1}$, instead of one inside $\mathbb{R}^d$. From this perspective, (Wang & Wang, 2022, Theorem 3.1) shows that (4) can be rewritten as an intrinsic SDE on $\mathbb{S}^{d-1}$

$$\mathrm{d}\overline{X}_t = -\gamma_t^{-\frac{1}{2}}\Big(\overline{\nabla}L(\overline{X}_t)\mathrm{d}t + \bar{\sigma}(\overline{X}_t)\mathrm{d}B_t^K\Big), \tag{6}$$

where $\bar{\sigma}(\cdot)^{\frac{1}{2}}$ is a tensor field along $\mathbb{S}^{d-1}$ whose value is given by the restriction of $\sigma(\cdot)^{\frac{1}{2}}$ and $\overline{\nabla}$ is the gradient operator on $\mathbb{S}^{d-1}$. (See the remark after (Wang & Wang, 2022, Theorem 3.1) for the

meaning of being intrinsic. In particular, $\overline{\nabla} L(\overline{X}_t)\mathrm{d}t$, $\sigma(\overline{X}_t)\mathrm{d}B_t^K$ are viewed as vector fields along $\mathbb{S}^{d-1}$.)

**Remark 2.1.** *Instead of the term $\bar{\sigma}(\overline{X}_t)\mathrm{d}B_t^K$ in (4) and (6), the papers (Li et al., 2020; Wang & Wang, 2022) actually used the restriction of $\Sigma(\overline{X}_t)^{\frac{1}{2}}\mathrm{d}B_t^d$ to $\mathbb{S}^{d-1}$. However, these two expressions are equivalent as Wiener processes because $\Sigma = \sigma\sigma^\top$.*

Since (6) is a perturbation with Brownian noise of the gradient flow $\mathrm{d}X_t = -\gamma^{-\frac{1}{2}}\overline{\nabla}L(\overline{X}_t)$ with varying learning rate $\gamma_t^{-\frac{1}{2}}$, it makes sense to first understand the constant speed gradient flow

$$\overline{X}_t = -\overline{\nabla}L(\overline{X}_t), \tag{7}$$

which is an ODE on the compact manifold $\mathbb{S}^{d-1}$. Following earlier notation, the local minima of $L$ on $\mathbb{S}^{d-1}$ is $\Gamma_1$ and has connected components $\Gamma_1^i$.

Write $U_1^i \in \mathbb{S}^{d-1}$ for the attracting basin of $\Gamma_1^i$, i.e. the set of $\overline{X}_0 \in \mathbb{S}^{d-1}$ such that the solution $\overline{X}_t$ to (7) with initial value $\overline{X}_0$ satisfies $\lim_{t\to\infty}\overline{X}_t \in \Gamma_1^i$).

**Lemma 2.2.** *Under Assumption 1.2.(ii), the $U_1^i$'s for $i = 1, \cdots, m$ are disjoint open sets of $\mathbb{S}^{d-1}$. Moreover, their union $\bigsqcup_{i=1}^m U_1^i$ has full volume in $\mathbb{S}^{d-1}$, and $\mathbb{S}^{d-1}\backslash\bigsqcup_{i=1}^m U_1^i$ is a proper submanifold of $\mathbb{S}^{d-1}$.*

The proof of Lemma 2.2 is standard and we left it to the reader. The key observation is that the complement $\mathbb{S}^{d-1}\backslash\bigsqcup_{i=1}^m U_1^i$ is the union of attracting basins of the connected components of critical points that are not local minima. By Assumption 1.2.(i), those critical points are saddle like and their attracting basins are proper submanifolds.

Note that $L$ is constant on $\Gamma_1^i$ and $\Gamma^i$, and $L(\Gamma_1^i) = L(\Gamma_1^i)$ is the minimum of $L$ inside $U_1^i$. For $q > 0$, write

$$U_1^{i,q} := \{x \in U_1^i : L(x) \le L(U_1^i) + q\}.$$

Define cones $U^i = \{x \in \mathbb{R}^d\backslash\{0\} : \frac{x}{|x|} \in U_1^i\}$, $U^{i,q} = \{x \in \mathbb{R}^d\backslash\{0\} : \frac{x}{|x|} \in U_1^{i,q}\}$. Then $U^i$ the attracting basin of $\Gamma^i$ under the gradient flow

$$X_t = -\nabla L(X_t). \tag{8}$$

By (Arora et al., 2022, Lemma B.15), under Assumption 1.2.(i), $U^i$ is open and the function $\Phi(x) = \lim_{t\to\infty}X_t$ is $C^2$-differentiable on $U^i$.

## 2.2 MAIN RESULT

**Definition 2.3.** *We define the Lipschitz distance between two probability measures $\mu$, $\nu$ on a metric space $X$ as*

$$\mathrm{dist}_{\mathrm{Lip}}(\mu, \nu) = \sup_{\|\varphi\|_{\mathrm{Lip}}\le 1}\Big|\int\varphi\mathrm{d}\mu - \int\varphi\mathrm{d}\nu\Big|,$$

*where the Lipschitz norm of a function $\varphi$ is given by $\|\varphi\|_{\mathrm{Lip}} := \max\left(\|\varphi\|_{C^0}, \sup_{x\ne y}\frac{|\varphi(x)-\varphi(y)|}{\mathrm{d}(x,y)}\right)$.*

We are now able to state our main theorem, which is a mutual reinforcement to both (Li et al., 2022c, Theorem 5.5) and (Wang & Wang, 2022, Theorem 4.6).

**Theorem 2.4.** *Under Assumption 1.2, for all $\epsilon > 0$ and compact interval $[\rho_-, \rho_+] \subset (0, \infty)$, there exist a constant $c > 0$ and a set $\Lambda \subseteq \bigsqcup_{i=1}^m U_1^i \subseteq \mathbb{S}^{d-1}$ of volume $\mathrm{vol}_{\mathbb{S}^{d-1}}(\Lambda) > 1 - \epsilon$, such that for all $\eta \le O(\lambda) \le O(1)$, the following holds:*

*For all initial parameter $x_0 \in \mathbb{R}^d$ with $|x_0| \in [\rho_-, \rho_+]$ and $\frac{x_0}{|x_0|} \in \Lambda$, all growth rates $K$ such that $K \to \infty$ as $\eta\lambda \to 0$, and all time values*

$$t \in \left[\frac{K\log(1 + \frac{\lambda}{\eta})}{\eta\lambda}, e^{\frac{c}{\sqrt{\eta\lambda}}}\right],$$

*the random trajectory to (2) with initial value $x_0$ satisfies*

$$\mathrm{dist}\big(\mathcal{P}_{X_0=x_0}(X_t), \nu^i\big) < \epsilon,$$

*where $\nu^i$ is a probability measure supported on the attractor $\Gamma^i$ of the unique attracting basin $U^i$ containing $x_0$, and $\nu^i$ only depend on $L$, $\sigma$ and $i$.*

Here $\mathrm{vol}_{\mathbb{S}^{d-1}}$ is the renormalized volume on the sphere $\mathbb{S}^{d-1}$ so that the total mass of $\mathbb{S}^{d-1}$ is 1.

## 3    REMOVAL OF ANALYTICITY ASSUMPTION

In this part, we will prove that Assumption 1.3.(ii) on analyticity (which is (Li et al., 2022c, Assumption 5.3)) is unnecessary for (Li et al., 2022c, Theorem 5.4), and thus (Li et al., 2022c, Theorem 1.2) holds without such an assumption as well.

**Proposition 3.1.** *Under Assumption 1.2 and Assumption 1.3.(i), the conclusion of (Li et al., 2022c, Theorem 1.2) hold.*

For this purpose, we temporarily adopt the setting of (Li et al., 2022c) for now. In other words, Assumption 1.3.(i) is being assumed and $\Gamma_1$ is a compact connected submanifold of $\mathbb{S}^{d-1}$ and $\Gamma = \{rx : r > 0; x \in \Gamma_1\}$. By (Li et al., 2022c, Theorem 4.1) it suffices to prove the (qualitative) mixing of the SDE (Li et al., 2022c, Equation (13)) on $\Gamma$ towards a unique invariant probability measure. Using the notations from (Li et al., 2022c, Chapter E.3), this SDE writes:

$$\mathrm{d}Y_t = f_0(Y_t)\mathrm{d}t + \sum_{k=1}^{K} f_k(Y_t) \cdot \mathrm{d}B_{k,t}, \tag{9}$$

where $f_k$ are certain vector fields along the radial cone $\Gamma \subseteq \mathbb{R}^d \backslash \{0\}$ and $B_{k,t}, k = 1, \cdots K$ are i.i.d. Wiener processes. Indeed, $f_1, \cdots, f_k$ are orthogonal to the radial direction with

$$f_k(x) = \partial \Phi(x)\sigma_k(x), 1 \le k \le K$$

and they span $T_x \Gamma \cap x^{\parallel} = T_x \Gamma_{|x|}$ for all $x \in \Gamma$, where $\Gamma_r = \{x \in \Gamma : |x| = r\}$, and $f_0$ has the form

$$f_0(x) = \frac{1}{2}\Big( - x + \partial^2 \Phi(x)[\Sigma(x)] - \frac{1}{2}\sum_{i=k}^{K} \partial f_k(x) f_k(x)\Big).$$

In the proofs from (Li et al., 2022c), analyticity is only used in Chapter F.4, when $\mathrm{Tr}\,\Sigma$ is not a constant on $\Gamma_1$. In this case, it was proved there (without using analyticity) that $\Gamma_* := \{y \in \Gamma : r_- \le |x| \le r_+\}$ with

$$r_- = \min_{\Gamma_1}(\mathrm{Tr}\,\Sigma)^{\frac{1}{4}}, r_+ = \max_{\Gamma_1}(\mathrm{Tr}\,\Sigma)^{\frac{1}{4}} \tag{10}$$

is the unique invariant control set for the control problem corresponding to (9).

Instead of using Kliemann's condition (Kliemann, 1987), which requires the vector field Lie algebra $\mathfrak{l}$ generated by $f_0, \cdots, f_N$ to be of maximal dimension at all points in $\Gamma_*$ and is the reason for the need of analyticity in (Li et al., 2022c), we will use Arnold-Kliemann's condition (Arnold & Kliemann, 1987), which only requires that the vector field Lie algebra $\mathfrak{l}$ to be of maximal dimensional at one point in $\Gamma_*$. This condition is true because the projection of $f_0$ to the radial direction is not constantly 0 if $\mathrm{Tr}\,\Sigma$ is not a constant. (Otherwise $\Gamma_*$ wouldn't be the unique invariant set.) Under this condition, (Arnold & Kliemann, 1987, Theorem 5.1) proved that there is a unique invariant probability measure $\nu$ supported on $\overline{\Gamma_*}$. The measure $\nu$ then has to be ergodic. Moreover, (Arnold & Kliemann, 1987, Theorem 5.2) showed that $\nu$ is absolutely continuous with respect to the Riemannian volume on the manifold $\Gamma$. In particular, $\nu(\partial \Gamma_*) = 0$ and $\nu(\Gamma_*) = 1$.

The main issue is to prove the convergence of the distribution $\mathcal{S}_{t,x}$ towards $\nu$ as $t \to \infty$, where $\mathcal{S}_{t,x}$ denotes the measure of all trajectories of solutions to (9) at time $t$ starting from $x \in \Gamma_*$. A priori, such convergence is only known to hold for $\frac{1}{T}\int_0^T \mathcal{S}_{t,x}\mathrm{d}t$ by ergodic theorem.

Applying (Duflo & Revuz, 1969, Theorem II.4) to $(\Gamma_*, \nu)$, it suffices to check two conditions to guarantee $\mathcal{S}_{t,x} \to \nu$ (in total variation distance):

**(Harris's recurrence condition)** For all sets $A$ with $\nu(A) > 0$ and all $x \in \Gamma_*$,

$$\mathbb{P}_{X_0=x}\left(\int_0^{\infty} 1_A(X_t)\mathrm{d}t = \infty\right) = 1; \tag{11}$$

**(Regularity condition)** The absolutely continuous component of $\mathcal{S}_{t,x}$ respect to $\nu$, called $\mathcal{S}_{t,x}^0$, satisfies $\lim_{t\to\infty} \mathcal{S}_{t,x}^0(\Gamma_*) = 1$ for all $x \in \Gamma_*$.

Let us first verify (**Harris's recurrence condition**). Define $D \subset \Gamma_*$ by

$$D = \{y \in \Gamma_* : f_0(x) \notin T_x \Gamma_{|x|}\}.$$

Then $D$ is open in $\Gamma_*$. For all $x \in \Gamma_*$ and $A \subseteq \Gamma_*$, we define the random variable $\tau_{x,A} \geq 0$ to be the first entering time into $A$ for a trajectory of (9) starting at $y$.

**Lemma 3.2.** $\mathbb{P}(\tau_{x,D} < \infty) = 1$ *for all* $x \in \Gamma_*$.

*Proof of* (**Harris's recurrence condition**). By (Arnold & Kliemann, 1987, Theorem 6.1), (11) holds for all $x \in D \cap \mathrm{int}\Gamma_* = D$. By Lemma 3.2, it then holds for all $x \in \Gamma_*$. This verifies Harris's recurrence condition. $\qquad\square$

We now verify the (**Regularity condition**): In addition to the Lie algebra $\mathfrak{l}$ and set $D$, define a Lie algebra $\mathfrak{l}_0 \subseteq T\Gamma$ and an open set $D_0$ by

$$\mathfrak{l}_0 = \mathrm{span}(f_1, \cdots, f_N) + [\mathfrak{l}, \mathfrak{l}], \ D_0 = \{y \in \Gamma_* : \dim \mathfrak{l}_0(x) = \dim \Gamma_*\}.$$

It is easy to see that $\mathfrak{l}_0$ is indeed a Lie algebra and $D_0$ is a relatively open subset in $\Gamma_*$

**Lemma 3.3.** *The set $D_0$ has non-empty interior.*

We postpone the proofs of Lemma 3.2 and 3.3 to Appendix A.

**Lemma 3.4.** *For all $x \in D_0$ and $t > 0$, $\mathcal{S}_{t,x}^0(\Gamma_*) > 0$.*

*Proof.* By (Ichihara & Kunita, 1974, Lemma 2.1), at every $x \in \mathrm{int}D_0$, the second order differential operator on the right hand side of (9) is elliptic (non-degenerate) at $x$. The lemma follows. $\qquad\square$

*Proof of* (**Regularity condition**). By (**Harris's recurrence condition**) and Lemma 3.3, for all $x \in \Gamma_*$, $\mathbb{P}(\tau_{x,D_0} < \infty) > 0$, and thus there exists $t_0(x) > 0$ such that $\mathbb{P}(\tau_{x,D_0} < t_0(x)) > 0$. In other words, on a subset $\Omega_{x,D_0} \subset \Omega$ of stochastic incidences $\omega$ with $\mathbb{P}(\Omega_{x,D_0}) > 0$, there exists $\tau'_{x,D_0} = \tau'_{x,D_0}(\omega) \in (0, t_0(x))$ such that the solution starting at $Y(0) = x$ satisfies $Y(\tau'_{x,D_0}) \in D_0$. By Lemma 3.4, for all $t > t_0(x)$, $\mathcal{S}_{t,x}^0(\Gamma_*) \geq \int_{\Omega_{x,D}} \mathcal{S}_{t-\tau'_{x,D_0},Y(\tau'_{x,D_0})}^0(\Gamma_*) > 0$. This shows that the statement "For $\nu$-a.e. $x$, $\mathcal{S}_{t,x}^0(\Gamma_*) = 0$ for all $t > 0$" is false. By (Duflo & Revuz, 1969, Proposition, p235), this guarantees (**Regularity condition**). $\qquad\square$

We have by now completed the proof of the mixing property $\mathcal{S}_{t,x} \to \nu$ under the generic Assumption 1.2, and the Assumption 1.3.(i).

## 4 REMOVAL OF UNIQUE BASIN ASSUMPTION

We now stop assuming Assumption 1.3.(i) and decompose $\Gamma = \bigsqcup \Gamma^i$ where each $\Gamma^i$ is a connected cone. Write $\Gamma_\# = \Gamma \cap \{|x| \in [R_-, R_+]\}$ where we fix $R_- < \left(\frac{\eta \min_{|x|=1} \mathrm{Tr}\, \Sigma(x)}{2\lambda}\right)^{\frac{1}{4}}$ and $R_+ > \left(\frac{\eta \max_{|x|=1} \mathrm{Tr}\, \Sigma(x)}{2\lambda}\right)^{\frac{1}{4}}$ near these bounds.

Note that $\Gamma \cap \{|x| \in [\left(\frac{\eta \min_{|x|=1} \mathrm{Tr}\, \Sigma(x)}{2\lambda}\right)^{\frac{1}{4}}, \left(\frac{\eta \max_{|x|=1} \mathrm{Tr}\, \Sigma(x)}{2\lambda}\right)^{\frac{1}{4}}]$ is an invariant control set of (2). Moreover, it was proved in (Li et al., 2020) that for a given initial radius $|x_0|$, the radius $|X_t|$ of (2) starting at $x_0$ will be almost surely inside $[R_-, R_+]$ for $t \geq O((\eta\lambda)^{-1})$.

Write $\Gamma_r^i = \Gamma^i \cap \{|x| = r\}$ and $\Gamma_\#^i = \Gamma^i \cap \Gamma_\#$. Then $\Gamma_1^i$ are the connected components of the manifold $\Gamma_1$. In particular, there are only finitely many $\Gamma_i$'s. Write $d^i = \dim \Gamma^i$. For $s > 0$, write $U_{1,p}^i$ for the $p$-neighborhood of $\Gamma_1^i$ in $\Gamma_1$, $U_{r,p}^i = rU_{1,p}^i$ and $U_{\#,p}^i = \bigcup_{r \in [R_-, R_+]} U_{r,p}^i$.

Fix from now on a sufficiently small parameter $p_0$ such that $U_{1,p_0}^i$ are disjoint for distinct $i$'s, and $\Gamma_1^i$ is the set of all critical points of $L$ inside $U_{1,10p_0}^i$. This is possible because of Assumption 1.2.(ii).

**Proposition 4.1.** *For all sufficiently small $p_1$ (to be determined later), given $K > 1$, $\epsilon > 0$, there exists a subset $\Lambda_{K,\epsilon} \subseteq \mathbb{S}^{d-1}$ with $\mathrm{m}_{\mathbb{S}^{d-1}}(\Lambda_{K,\epsilon}) > 1 - \epsilon$ such that for all $x_0$ with $|x_0| \in [\frac{1}{K}, K]$ and $\frac{x_0}{|x_0|} \in \Lambda_{K,\epsilon}$, with probability $> 1 - \epsilon$, the trajectory of (2) starting at $x_0$ will remain inside $\bigcup_i U_{\#,p_1}^i$ for $t \in [0, C_{\mathrm{des}}(\eta\lambda)^{-1}]$ for some constant $C_{\mathrm{des}}$.*

The proof of the proposition, which we omit, is a simple combination of (Li et al., 2020, Equation (7)) and the proof of (Wang & Wang, 2022, Theorem 4.5).

The proposition allows us to assume that our initial point is inside one of finitely many basins $U^i_{\#,p_1}$. To prove the main result, it now suffices to make two observations: First, with very high probability, the trajectory will not escape from the basin in exponential time. Second, as long as the trajectory remains in the basin, its distribution always mixes towards a unique probability measure $\nu^i$ supported at the bottom $\Gamma^i$ of the basin.

The first property is stated as Proposition 4.2 below.

**Proposition 4.2.** *There exist $C > 0$, and sufficiently small $p_0 > p_1 > 0$, such that for all $i$ and $x_0 \in U^i_{\#,p_1}$, in the regime $\eta \leq O(\lambda) \leq O(1)$, the solutions $X_t$ to (2) with initial position $x_0$ satisfy*

$$\lim_{\lambda\eta \to 0} \mathbb{P}_{X_0 = x_0}(X_t \text{ remains in } U^i_{\#,p_0} \text{ for } t \in [0, e^{C(\eta\lambda)^{-\frac{1}{2}}}]) = 1.$$

*The convergence is uniform with respect to the inital data $x_0$.*

It is similar in nature to (Wang & Wang, 2022, Lemma E.6) but requires a more sophisticated proof. This is the main theoretical component of this paper. The argument will be based on the large deviation principle of Dembo-Zeitouni in Dembo & Zeitouni (2010, Chapter 5), which was an adaptation of (Freidlin & Wentzell, 2012, Chapter 6). The reason for which Freidlin-Wentzell's original theory cannot be applied here like in (Wang & Wang, 2022) is that the diffusion in the SDE system (44), (45) is degenerate. Dembo-Zeitouni's work allows degenerate diffusions. However, further modifications to (Dembo & Zeitouni, 2010) are needed in our case as the first order drift in (45) depends on the $\gamma_t = |X_t|^4 \eta^{-2}$. We will treat $\gamma_t$ as a control variable. The full proof will be in Appendix B.

The second property follows from the main results (Theorem 5.1 & Theorem 6.7) in (Li et al., 2022c)) and is restated as Proposition 4.3 here with an additional emphasis on uniformity. A more detailed discussion can be found in Appendix C. Recall that (Li et al., 2022c)) also assumes Assumption 1.3.(ii), but that can be dropped by the discussion in Chapter 3 above.

**Proposition 4.3.** *Under Assumption 1.2 For all $K > 0$ and sufficiently small $p_0 > p_1 > 0$, such that in the regime $\eta \leq O(\lambda) \leq O(1)$ and $\eta\lambda \to 0$, the following holds: For each index $i$ and for all initial parameter $x_0 \in U^i_{\#,p_1}$, the distribution of all trajectories $\left\{X_t\right\}_{0 \leq t \leq \frac{K \log(1 + \frac{\lambda}{\eta})}{\eta\lambda}}$ of (2) that do not leave $U^i_{\#,p_1}$ converges in distribution to the trajectories $\{\hat{X}_t\}$ to a fixed SDE model (the Katzenberger model) supported on $\Gamma^i$ with initial position $\Phi(x_0)$. The convergence is uniform in $x_0$. Moreover, as $K \to \infty$ the trajectories $\{\hat{X}_t\}$ are uniformly mixing (with respect to different $x_0$'s) towards a fixed equilibrium measure $\nu^i$.*

Our main theorem, Theorem 2.4, then follows from combining Propositions 4.1, 4.2 and 4.3.

*Proof of Theorem 2.4.* By Proposition 4.1, after ignoring $O(\frac{1}{\eta\lambda})$ time at the beginning, as well as an $o(1)$ portion of stochastic incidences. One may assume $x_0 \in U^i_{\#,p}$ for some $i$. For $t$ within the range in the statement of Theorem 2.4, again after ignoring an $o(1)$ portion of incidences, one may assume that all trajectories under consideration stay within $U^i_{\#,p}$ up to time $t$. Because of the lower bound for $t$, one may consider a window of length $\frac{K \log(1 + \frac{\lambda}{\eta})}{\eta\lambda}$ that ends at $t$ where $K \to \infty$. The distribution of trajectories along this window is the average of distributions over different initial positions. By Proposition 4.3, all such components uniformly mix toward $\nu^i$. The proof is completed. $\square$

## 5 EXPERIMENTS

**Mixing on local manifold:** The key technical observation of this paper (Proposition 4.2) is that the distribution of trajectories with an initial position is trapped locally in the attracting basin containing the initial position during any practical observation windows. Using the method from (Wang & Wang, 2022, Fig. 13), this observation is supported by the experiment below: using a reduced MNIST dataset with only 1280 samples and a small CNN with 1786 parameters (so that the model is still overparametrized), we ran 15 independent instances of SGD, at $\lambda = \eta = \frac{1}{32}$, for each of two

randomly chosen initial parametrizations. Each instance lasts 0.8 million steps of SGD. A smilar experiment was ran for reduced CIFAR10 dataset with 1280 samples, a CNN model with 2658 parameters, $\eta = \frac{1}{1024}$, $\lambda = \frac{1}{32}$ and 1.28 million SGD steps. In order to show that the distribution arising from each initial position does stabilize toward an equilibrium and the two equilibria are different, we track the variance within each group, and compare them with the average distance square over all pairs of point s from different groups. Namely, denoting by $\{X_{i,t}^k\}$ the $i$-th trajectory starting at initial point $x_k$ where $k = 1, 2$, we compute the following quantities:

$$V_{11}(t) = \mathbb{E}_{i \neq j}|X_{i,t}^1 - X_{j,t}^1|^2, \ V_{22}(t) = \mathbb{E}_{i \neq j}|X_{i,t}^2 - X_{j,t}^2|^2, V_{12}(t) = \mathbb{E}_{i,j}|X_{i,t}^1 - X_{j,t}^2|^2.$$

Figures 1, 2 shows that $V_{11}$ and $V_{22}$ stabilizes near similar but different values, but $V_{12}$ stabilizes at a much bigger value. This suggests that the distributions of trajectories with starting point $x_1$ and $x_2$ mixes towards equilibria whose support have similar scales, but these two equilibria are far apart from each other.

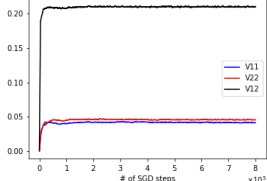

Figure 1: small MNIST: Variance comparison between distributions with different initial positions

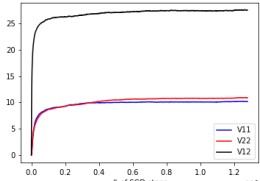

Figure 2: small CIFAR: Variance comparison between distributions with different initial positions

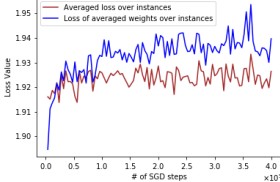

Figure 3: small MNIST: Comparison between training losses before and after SWAP

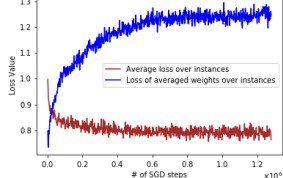

Figure 4: small CIFAR: Comparison between training losses before and after SWAP

**Prediction on the failure of stochastic weight averaging in parallel (SWAP):** Our theory predicts that if the local minima manifold of minimizes $\Gamma^i$ has non-trivial geometry, that is, the average of parameters on the manifold may fall off the manifold, then it might fail to decrease loss, or even increase the loss, once the SGD mixes to the local manifold.

We apply stochastic weight average over trajectories (SWAP) to the neural network parameters at each step over the 15 independent instances with the same initial position and compute the loss function at the averaged parameter. SWAP, a variant of stochastic weight average (SWA) from Izmailov et al. (2018), was proposed by Gupta et al. (2020). Figures 3, 4 show that although the loss of the SWAP parameter improves the average loss over the independent instances at the beginning, the improvement quickly breaks after a couple thousands of training steps. This phenomenon verifies our theoretical prediction and also suggests that the support of the equilibrium is not a convex set but rather a manifold of curved shape.

## 6 CONCLUSION

We give rigorous proof of fast equilibrium conjecture in generic situations, removing previous assumptions that there is only one basin and the set of minima is analytic. The main technical contribution is that we justify most of the trajectories of SDE would not escape from one basin within exponential time. Instead of using spectral analysis, we adopt the large deviation principle type of argument. Possible interesting direction may include understanding the dependence of mixing time on dimension, architecture and noise structure.

## 7 ACKNOWLEDGEMENT

Y.W. and Z.W. acknowledge respectively supports from NSF.

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

## A    PROOFS FOR SECTION 3

*Proof.* of Lemma 3.2. Instead of (9), define

$$\mathrm{d}Y_t^{\parallel} = f_0^{\parallel}(Y_t^{\parallel})\mathrm{d}t + \nabla_{i=1}^N f_i(Y_t^{\parallel}) \cdot \mathrm{d}B_{i,t}, \tag{12}$$

where $f_0^{\parallel}(x)$ is the projection of $f_0(x)$ to $T_x\Gamma_{|x|}$.

For all $0 \le t < \tau_{x,D}$, $f_0 = f_0^{\parallel}$ at $Y_t$, thus the solutions $Y$ and $Y^{\parallel}$ to (9) and (12) starting at $y$ coincide up to $\tau_{x,D}$. Moreover, the trajectories of (12) stay in $\Gamma_{|x|}$. So $\tau_{x,D}$ is also the first entering time into $D \cap \Gamma_{|x|}$ for (12). We claim that $D \cap \Gamma_{|x|}$ is a non-empty subset of $\Gamma_{|x|}$. Indeed, if this is not true, then the trajectories $Y$ and $Y^{\parallel}$ coincide forever and stay inside $\Gamma_{|x|}$, making $\Gamma_{|x|}$ an invariant control set which contradicts to the uniqueness of $\Gamma_*$. Moreover, $D \cap \Gamma_{|x|}$ is relatively open in the submanifold $\Gamma_{|x|}$. By Assumption 1.2.(iii), (12) has non-degenerate diffusion on the submanifold $\Gamma_{|x|} = |x|\Gamma_1$. Thus $Y^{\perp}$ enters $D \cap \Gamma_{|x|}$ almost surely. This proves the lemma.    □

*Proof.* of Lemma 3.4. The proof is similar to that of (Li et al., 2022c, Lemma F.18). Recall that $\mathrm{Tr}\,\Sigma$ is $(-2)$-homogeneous and assumed to be non-constant on $\Gamma_1$. Therefore, there is an open set $V_1 \in \Gamma_1$ and a vector field $f^*$ taking values in $\mathrm{span}(f_1, \cdots, f_N)$ such that $\langle \nabla \mathrm{Tr}\,\Sigma, f^* \rangle \ne 0$ on $V_1$. By homogeneity, this is also true on $rV_1 \subseteq \Gamma_r$ for all $r > 0$. As $\mathrm{Tr}\,\Sigma(x) = -\frac{1}{2}\langle x, f_0(x) + \frac{x}{2} \rangle$ and $\langle x, f^* \rangle = 0$, this implies $\langle \nabla \langle x, f_0(x) \rangle, f^* \rangle \ne 0$ on $rV_1$. By (Li et al., 2022c, Lemma F.16), $\langle x, [f_0, f^*] \rangle \ne 0$ on $rV_1$. Note that $[f_0, f^*] \in \mathfrak{l}_0$, and $\mathrm{span}(f_1, \cdots, f_N) = T\Gamma_r$ on $rV_1$ by controllability. It follows that $\{rV_1 : r \in (r_-, r_+)\} \subseteq D_0$, which is sufficient to conclude.    □

## B    EXITING TIME WITH DEGENERATE DIFFUSION AND EXTRA CONTROL VARIABLE

In this appendix, we establish a probabilistic lower bound to the exiting time of a stochastic process from a basin (Theorem B.21), based on a one-sided large deviation principle (Theorem B.12). Our proofs adapt those from the work of Dembo-Zeitouni (Dembo & Zeitouni, 2010, Chapter 5) to a more general setting. The main differences in the setting are:

1. The stochastic process in the basin is now governed not only by the current location and an Brownian motion, but also by an extra control variable as stated in equation (17);

2. The local minima set in the basin is no longer assumed to be a unique isolated fixed point.

The diffusion in the stochastic process is allowed to be degenerate, which was the main novelty in the Dembo-Zeitouni theory compared to the earlier work of Freidlin-Wentzell (Freidlin & Wentzell, 2012, Chapter 6).

*This appendix will solely consist of mathematical analysis. With the exception of §B.4, all notations are chosen independently from those used in other parts of the current paper.*

### B.1 PROPERTIES OF UPPER LARGE DEVIATION PRINCIPLE

In this section we define the notion of upper large deviation principle (upper LDP). This is the upper bound part of the large deviation principle defined in (Dembo & Zeitouni, 2010, Chapter 1.2). The principles proved in (Dembo & Zeitouni, 2010, Chapter 4.1 & 4.2), which allows to pass the LDP property between random processes, are still valid for upper LDP because the upper and lower bounds are treated separately in their proofs. The purpose of this section is to briefly list which facts are relevant and justify the survival of their proofs in (Dembo & Zeitouni, 2010) with upper LDP.

**Definition B.1.** *A rate function $I$ is a lower semicontinuous mapping $I : \mathcal{X} \to [0, \infty]$ on a metric space $\mathcal{X}$, i.e. $I^{-1}([0, a])$ is closed for all finite $a$. A good rate function is a rate function $I$ such that $I^{-1}([0, a])$ is compact for all finite $a$. The effective domain $\mathcal{D}_I$ if $I$ is $I^{-1}([0, \infty))$.*

**Definition B.2.** *A family of Borel probability measures $\{\mu^\epsilon\}$ on $(\mathcal{X}, \mathcal{B})$ satisfies upper large deviation principle (upper LDP) with rate function $I$ if for all measurable subsets $A$ of $\mathcal{X}$,*

$$\limsup_{\epsilon \to 0} \epsilon \log \mu^\epsilon(A) \leq - \inf_{x \in \overline{A}} I(x). \tag{13}$$

*A family of Borel probability measures $\mu^\epsilon$ on $(\mathcal{X}, \mathcal{B})$ satisfies weak upper large deviation principle (weak upper LDP) with rate function $I$ if (13) holds for all compact subsets $A$.*

For background, recall that $\{\mu^\epsilon\}$ is said to satisfy the large deviation principle with rate function $I$ if in addition to (13) it also satisfies the lower bound

$$\liminf_{\epsilon \to 0} \epsilon \log \mu^\epsilon(A) \geq - \inf_{x \in A^\circ} I(x). \tag{14}$$

**Theorem B.3.** *Suppose $\mathcal{A}$ is a base for the topology of $\mathcal{X}$. Then a family of probability measures $\{\mu^\epsilon\}$ on $\mathcal{X}$ satisfies weak upper LDP with rate function*

$$I(x) := \sup_{A \in \mathcal{A}, x \in A} \left( - \limsup_{\epsilon \to 0} \epsilon \log \mu^\epsilon(A) \right).$$

**Theorem B.4.** *(Contraction Principle) If $F : \mathcal{X} \to \mathcal{Y}$ is a continuous map between Hausdorff spaces and $I : \mathcal{X} \to [0, \infty]$ is a good rate function. Then*

   *(a) The function $I'(y) := \inf_{F^{-1}(\{y\})} I$ is a good rate function on $\mathcal{Y}$;*

   *(b) If a family of probability measures $\{\mu^\epsilon\}$ on $\mathcal{X}$ satisfies upper LDP with rate function $I$, then the pushforward measures $\{\mu^\epsilon \circ F^{-1}\}$ satisfies upper LDP with rate function $I'$ on $\mathcal{Y}$.*

*Remark about the proofs.* Theorem B.3 and Theorem B.4 are the upper bound directions of (Dembo & Zeitouni, 2010, Theorem 4.1.11 & 4.2.1). Their proofs are identical to those therein. For Theorem B.3, note that this direction only uses the equality (4.1.14), but not (4.1.12) and (4.1.13), in (Dembo & Zeitouni, 2010). □

**Definition B.5.** *For families $\{\nu^{\epsilon,m}\}$ and $\{\nu^\epsilon\}$ of probability measures on a metric space $\mathcal{Y}$, where $m \in \mathbb{N}$ and $\epsilon > 0$, we say $\{\nu^{\epsilon,m}\}$ are exponentially good approximations of $\{\nu^\epsilon\}$ if there exist probability spaces $(\Omega, \mathcal{B}^\epsilon, \mathcal{P}^{\epsilon,m})$ and two families of random variables $y^{\epsilon,m}, y^\epsilon$ with joint distribution $\mathcal{P}^\epsilon$ and marginal distributions $\nu^{\epsilon,m}, \nu^\epsilon$ such that for all $\delta > 0$, the event $\mathrm{dist}(y^{\epsilon,m}, y^\epsilon) > \delta$ is $\mathcal{B}^\epsilon$-measurable and*

$$\lim_{m \to \infty} \limsup_{\epsilon \to 0} \epsilon \log \mathcal{P}^\epsilon \Big( \mathrm{dist}(y^{\epsilon,m}, y^\epsilon) > \delta \Big) = -\infty. \tag{15}$$

*If in addition $\nu^{\epsilon,m} = \widetilde{\nu}^\epsilon$ is independent of $m$, we say $\{\widetilde{\nu}^\epsilon\}$ and $\{\nu^\epsilon\}$ are exponentially equivalent.*

**Theorem B.6.** *For families $\{\nu^{\epsilon,m}\}$ and $\{\nu^\epsilon\}$ of probability measures on a metric space $\mathcal{Y}$, where $m \in \mathbb{N}$ and $\epsilon > 0$, and $\{\nu^{\epsilon,m}\}$ are exponentially good approximations of $\{\nu^\epsilon\}$. If for each $m$, $\{\nu^{\epsilon,m}\}$ satisfies upper LDP with rate function $I_m$, then*

   *(a) $\{\widetilde{\nu}^\epsilon\}$ satisfies weak upper LDP with rate function*

$$I(y) := \sup_{\delta > 0} \limsup_{m \to \infty} \inf_{z \in B_\delta(y)} I_m(z).$$

*(b) If in addition $I$ is a good rate function and for every closed subset $A \subseteq \mathcal{Y}$,*

$$\inf_{Y \in A} I(Y) \leq \limsup_{m \to \infty} \inf_{Y \in A} I_m(Y).$$

*Remark about the proof.* The proof is the same as that of (Dembo & Zeitouni, 2010, Theorem 4.2.16). For part (a), by Theorem 3.3 applied to the topological base consisting of all metric balls $B_\delta(y)$ in $\mathcal{Y}$, $\{\nu^{\epsilon,m}\}$ satisfies weak upper LDP with rate function

$$I^*(y) := \sup_{\delta > 0} \big( -\limsup_{\epsilon \to 0} \epsilon \log \nu^\epsilon(B_\delta(y)) \big).$$

So it suffices to prove $I(y) \leq I^*(y)$. This was done in the proof of (Dembo & Zeitouni, 2010, Theorem 4.2.16, part (a)) via the inequality

$$\limsup_{\epsilon \to 0} \epsilon \log \nu^\epsilon(B_\delta(y)) \leq \liminf_{m \to \infty} \big( -\inf_{z \in \overline{B_{2\delta}(y)}} I_m(z) \big) = -\limsup_{m \to \infty} \inf_{z \in \overline{B_{2\delta}(y)}} I_m(z).$$

Hence

$$I^*(y) = \sup_{\delta > 0} \big( -\limsup_{\epsilon \to 0} \epsilon \log \nu^\epsilon(B_\delta(y)) \big) \geq \sup_{\delta > 0} \limsup_{m \to \infty} \inf_{z \in \overline{B_{2\delta}(y)}} I_m(z) = I(y).$$

The proof of part (b) is verbatim as in (Dembo & Zeitouni, 2010). $\qquad\square$

**Theorem B.7.** *Suppose a family of probability measures $\{\mu^\epsilon\}$ satisfies upper LDP with a good rate function $I$ on a Hausdorff topological space $\mathcal{X}$. And suppose a sequence of continuous maps $\{F^m\}$ from $\mathcal{X}$ to another Hausdorff topological space $\mathcal{Y}$ approximate a measurable maps $F$ in the sense that for all $a < \infty$,*

$$\limsup_{m \to \infty} \sup_{\{x : I(x) \leq a\}} \operatorname{dist}(F^m(x), F(x)) = 0.$$

*Finally, assume the families $\{\mu^\epsilon \circ (F^m)^{-1}\}$ are exponentially good approximations of another family of probability distributions $\{\nu^\epsilon\}$ on $\mathcal{Y}$. Then $\{\nu^\epsilon\}$ satisfies upper LDP with good rate functions $I'(y) := \inf_{F^{-1}(\{y\})} I$.*

*Remark about the proof.* The theorem is the upper bound part of of (Dembo & Zeitouni, 2010, Theorem 4.2.23). The proof stay the same with (Dembo & Zeitouni, 2010, Theorems 4.2.1 & 4.2.16) replaced by their respective upper bounds direction, namely Theorem B.4 and Theorem B.6. $\qquad\square$

### B.2 Upper LDP for degenerate diffusion with extra control variable

In this part, we prove a variation of the upper bound direction in the large deviation principle proved in (Dembo & Zeitouni, 2010, Theorem 5.6.7). The notations in this section are self-contained and independent of those from other parts of this paper.

Throughout this section we will consider the following setting:

- $U \subseteq \mathbb{R}^n$ is an open domain,
- $D \subseteq \mathbb{R}^l$ be a compact set,
- $b : U \to \mathbb{R}^n$, $\sigma : U \times D \to L(\mathbb{R}^d, \mathbb{R}^n)$ and $h : \mathbb{R} \times U \times D \to \mathbb{R}^l$ are Lipschitz continuous functions,
- $b$ and $\sigma$ are bounded, and $h$ is bounded on $[0, \epsilon_0] \times U \times D$ for some $\epsilon_0 > 0$. We will fix $\epsilon_0$ and a common upper bound $H$ for $b$, $\sigma$ and $h$ respectively on these domains.

Consider the following families of stochastic differential equations on $(Y, Z) \in U \times \mathbb{R}^l$ and $\epsilon > 0$:

$$\mathrm{d}Y_t^\epsilon = b(Y_t^\epsilon, Z_t^\epsilon)\mathrm{d}t + \sqrt{\epsilon}\sigma(Y_t^\epsilon, Z_t^\epsilon)\mathrm{d}B_t^d; \tag{16}$$

$$\mathrm{d}Z_t^\epsilon = h(\epsilon, Y_t^\epsilon, Z_t^\epsilon)\mathrm{d}t. \tag{17}$$

The main difference of our setting from that in (Dembo & Zeitouni, 2010) is the existence of the additional control variable $Z_t^\epsilon$, whose evolution depends on $\epsilon$ in a less prescribed way.

We will assume throughout this section, in addition to Assumption B.8, that:

**Assumption B.8.** *For all $\epsilon > 0$ and initial values $(y_0, z_0) \in U \times D$, the solution $(Y_t^\epsilon, Z_t^\epsilon)$ to (16) and (17) starting at $(y_0, z_0)$ almost surely remains in $U \times D$ for all $t > 0$.*

**Definition B.9.** *Given the functions $b$, $\sigma$, the upper bound $H$ on $|h|$, and $T > 0$. The associated path space $\mathcal{S}_T$ is defined as the family of triples $(f, g, u)$ with $f \in C^0([0,T], U)$, $g \in C_H^{\mathrm{Lip}}([0,T], D)$ and $u \in W^{1,2}([0,T], \mathbb{R}^d)$ that*

$$f_t = y_0 + \int_0^t b(f_s, g_s)\mathrm{d}s + \int_0^t \sigma(f_s, g_s)\mathrm{d}u_s. \tag{18}$$

Here $C_H^{\mathrm{Lip}}([0,T], D)$ is the subspace in $C^0([0,T], D)$ of functions $g$ with Lipschitz constant bounded by $H$, i.e. that satisfy

$$\sup_{s \neq t} \left| \frac{g(s) - g(t)}{s - t} \right| \leq H. \tag{19}$$

And $W^{1,2}$ is the square integrable Sobolev space of first order differentiability.

**Lemma B.10.** $C_H^{\mathrm{Lip}}([0,T], D)$ *is compact in $C^0([0.T], D)$.*

*Proof.* As $C^0([0.T], D)$ is a metric space, it suffices to show any sequence $g^{(k)}$ in $C_H^{\mathrm{Lip}}([0,T], D)$ has a convergent subsequence with limit in $C_H^{\mathrm{Lip}}([0,T], D)$

Because $D$ is a bounded domain, the $g^{(k)}$'s are uniformly bounded. As they are also uniformly Lipschitz with Lipschitz constant bounded by $H$, by Arzelà-Ascoli Theorem we may assume $g^{(k)}$ converges in $C^0$ to some $g \in C^0([0,T], D)$. Then $g$ would be $H$-Lipschitz continuous as well. $\square$

From now on, $C_H^{\mathrm{Lip}}([0,T], D)$ will be equipped with the $C^0$ topology without further notice.

**Definition B.11.** *Given a function $f \in C^0([0,T], U)$, the corresponding energy functional is*

$$\Phi_T(f) := \inf_{\substack{(g,u) \text{ such that} \\ (f,g,u) \in \mathcal{S}_T}} \int_0^T \frac{1}{2}|\dot{u}_t|^2 \mathrm{d}t.$$

*By convention, $\Phi_T(f) = \infty$ if $\mathcal{S}_T$ is empty.*

**Theorem B.12.** *Given a closed subset $\mathcal{A}$ of the metric space $\mathcal{A} \subset C^0([0,T], U)$ and an initial value $y_0$, for the solution $(Y_t^\epsilon, Z_t^\epsilon)$ to (16) and (17) with initial value $(y_0, z_0)$, the following inequality holds:*

$$\limsup_{\epsilon \to 0} \epsilon \sup_{z_0 \in D} \log \mathbb{P}_{(Y_0^\epsilon, Z_0^\epsilon) = (y_0, z_0)}(Y_t^\epsilon \in \mathcal{A}) \leq - \inf_{\substack{f \in \mathcal{A} \\ f_0 = y_0}} \Phi_T(f).$$

In order to prove Theorem B.12, we need more notations to study the $Y_t^\epsilon$ while keeping the path $Z_t^\epsilon$ fixed. For this purpose, we define some distributions of $(Z_t^\epsilon, \sqrt{\epsilon}B_t^d)$ and $Y_t^\epsilon$ respectively.

**Definition B.13.** *Denote by $\mu_{(y_0, z_0), T}^\epsilon$ the joint distribution of $(Z_t^\epsilon, \sqrt{\epsilon}B_t^d)$ on $C_H^{\mathrm{Lip}}([0,T], D) \times C^0([0,T], \mathbb{R}^d)$, and $\nu_{(y_0, z_0), T}^\epsilon$ for the distribution of $Y_t^\epsilon$ in $C^0([0,T], U)$, where $(Y_t^\epsilon, Z_t^\epsilon)$ is solution to (16), (17) with initial value $(y_0, z_0)$ at $t = 0$. Write $\lambda^\epsilon$ for the distribution of $\sqrt{\epsilon}B_t^d$ on $C^0([0,T], \mathbb{R}^d)$.*

**Definition B.14.** *Denote by $\mathcal{M}_T^\epsilon$ the space of all probability distributions $\mu$ on $C_H^{\mathrm{Lip}}([0,T], D) \times C^0([0,T], \mathbb{R}^d)$ whose projection to the second coordinate is $\lambda^\epsilon$.*

For all initial values $(y_0, z_0) \in U \times D$, $Y_t^\epsilon$, $Z_t^\epsilon$ are progressively measurable processes. In consequence,

$$\mu_{(y_0, z_0), T}^\epsilon \in \mathcal{M}_T^\epsilon. \tag{20}$$

**Lemma B.15.** *The function*

$$I(g, u) := \begin{cases} \displaystyle\int_0^1 \frac{1}{2}|\dot{u}_t|^2 \mathrm{d}t & \text{if } u \in W^{1,2}([0,1], \mathbb{R}^d) \\ \infty & \text{otherwise} \end{cases}$$

*is a good rate function on $C_H^{\mathrm{Lip}}([0,1],D) \times C^0([0,1],\mathbb{R}^d)$. Moreover, any family of probability measures $\{\mu^\epsilon\}$ on $C_H^{\mathrm{Lip}}([0,1],D) \times C^0([0,1],\mathbb{R}^d)$ such that $\mu^\epsilon \in \mathcal{M}_1^\epsilon$ satisfies upper LDP with rate function $I$.*

*Proof.* We first check that $I$ is a rate function. That is, it is lower semicountinuous on $C_H^{\mathrm{Lip}}([0,1],D) \times C^0([0,1],\mathbb{R}^d)$. It suffices to show that that $I(g,u) \leq \liminf_{k\to\infty} I(g^{(k)},u^{(k)})$ if $(g^{(k)},u^{(k)}) \to (g,u)$ in $C^0$ norm.

By Lemma B.10, $g$ is in $C_H^{\mathrm{Lip}}([0,1],D)$, and thus by definition $I(g,u) = I_0(u)$ in this case.

It was known by Schilder's Theorem ((Dembo & Zeitouni, 2010, Theorem 5.2.3)) that the function

$$I_0(u) := \begin{cases} \int_0^1 \frac{1}{2}|\dot{u}_t|^2 \mathrm{d}t & \text{if } u \in W^{1,2}([0,1],\mathbb{R}^d); \\ \infty & \text{otherwise.} \end{cases}$$

is a good rate function, and in particular lower semicountinuous. Thus

$$\liminf_{k\to\infty} I(g^{(k)},u^{(k)}) = \liminf_{k\to\infty} I_0(u^{(k)}) \geq I_0(u) = I(g,u).$$

Thus $I$ is a good rate function and we want to show that it is good, i.e. $I^{-1}([0,a])$ is compact for all finite $a$. Remark that $I^{-1}([0,a]) = C_H^{\mathrm{Lip}}([0,1],D) \times I_0^{-1}([0,a])$ and the second factor is compact as $I_0$ is a good rate function. Note that $I_0(u) = I(g,u)$. So it suffices to know that $C_H^{\mathrm{Lip}}([0,1],D)$ is a compact space in $C^0$ topology, which is the assertion of Lemma B.10.

Finally we need to show that (13) holds for $\{\mu^\epsilon\}$ and $I$. The same inequality holds for $\{\lambda^\epsilon\}$ and $I_0$, i.e. for all $\mathcal{A}_0 \subseteq C^0([0,1],\mathbb{R}^d)$,

$$\limsup_{\epsilon\to0} \epsilon \log \lambda^\epsilon(\mathcal{A}_0) \leq - \inf_{u\in\overline{\mathcal{A}_0}} I_0(u).$$

For all measurable set $\mathcal{A} \subset C_H^{\mathrm{Lip}}([0,1],D) \times C^0([0,1],\mathbb{R}^d)$, let $\mathcal{A}_0$ denote its projection to $C^0([0,1],\mathbb{R}^d)$. Then

$$\begin{aligned} \limsup_{\epsilon\to0} \epsilon \log \mu^\epsilon(\mathcal{A}) &\leq \limsup_{\epsilon\to0} \epsilon \log \mu^\epsilon\big(C_H^{\mathrm{Lip}}([0,1],D) \times \mathcal{A}_0\big) \\ &= \limsup_{\epsilon\to0} \epsilon \log \lambda^\epsilon(\mathcal{A}_0) \\ &\leq - \inf_{u\in\overline{\mathcal{A}_0}} I_0(u) \leq - \inf_{(g,u)\in\overline{\mathcal{A}}} I(g,u). \end{aligned}$$

In the last inequality, we used the fact that $\overline{\mathcal{A}_0}$ contains the projection of $\overline{\mathcal{A}}$ and that $I(g,u) = I_0(u)$. This completes the proof. $\square$

For all $(g,u) \in C_H^{\mathrm{Lip}}([0,1],D) \times C^0([0,1],\mathbb{R}^d)$, denote by $Y_{(g,u),t}^\epsilon$ the solution on $[0,1]$, with initial value $Y_{(g,u),0}^\epsilon = y_0$ to the stochastic differential equation

$$\mathrm{d}Y_{(g,u),t}^\epsilon = b(Y_{(g,u),t}^\epsilon, g_t)\mathrm{d}t + \sigma(Y_{(g,u),t}^\epsilon, g_t)\mathrm{d}u_t. \tag{21}$$

In addition, given an integer $m \in \mathbb{N}$, we also define $Y_{(g,u),t}^{\epsilon,m}$ as the solution on $[0,1]$, also with initial value $y_0$, to the following stochastic differential equation

$$\mathrm{d}Y_{(g,u),t}^{\epsilon,m} = b(Y_{(g,u),\frac{[mt]}{m}}^{\epsilon,m}, g_t)\mathrm{d}t + \sigma(Y_{(g,u),\frac{[mt]}{m}}^{\epsilon,m}, g_t)\mathrm{d}u_t. \tag{22}$$

We emphasize that $Y_{(g,u),t}^\epsilon$ and $Y_{(g,u),t}^{\epsilon,m}$ are deterministic once the pair $(g,u)$ are given and all randomness comes from $\mu_{(y_0,z_0),T}^\epsilon$.

**Lemma B.16.** *For all any $\delta > 0$ and all initial values $y_0 \in D$,*

$$\lim_{m\to\infty} \limsup_{\epsilon\to0} \epsilon \sup_{\mu\in\mathcal{M}_1^\epsilon} \log \mathbb{P}_{(g,u)\sim\mu}\big( \sup_{t\in[0,1]} |Y_{(g,u),t}^{\epsilon,m} - Y_{(g,u),t}^\epsilon| \geq \delta\big) = -\infty.$$

*Proof of Lemma B.16.* Fix $\delta > 0$. Let $\Delta_{(g,u),t}^{\epsilon,m} := Y_{(g,u),t}^{\epsilon,m} - Y_{(g,u),t}^{\epsilon}$ , for any $\rho > 0$, define the stopping time

$$\tau^{\epsilon,m,\rho} := \min(\inf\{t : |Y_{(g,u),t}^{\epsilon,m} - Y_{(g,u),\frac{[mt]}{m}}^{\epsilon,m}| \geq \rho\}, 1).$$

$\tau^{\epsilon,m,\rho}$ depends on $(g,u)$, but we skip it to simplify the notation.

The process $\Delta_{(g,u),t}^{\epsilon,n}$ satisfies the SDE

$$d\Delta_{(g,u),t}^{\epsilon,n} = b_t^{\Delta} dt + \sqrt{\epsilon}\sigma_t^{\Delta} du_t$$

with coefficients

$$b_t^{\Delta} := b(Y_{(g,u),\frac{[mt]}{m}}^{\epsilon,m}, g_t) - b(Y_{(g,u),t}^{\epsilon}, g_t),$$

$$\sigma_t^{\Delta} := \sigma(Y_{(g,u),\frac{[mt]}{m}}^{\epsilon,m}, g_t) - \sigma(Y_{(g,u),t}^{\epsilon}, g_t).$$

By the uniform Lipschitz continuity of $b$ and $\sigma$, and the boundedness of $g_t \in D$, there is a constant $C$ such that for all $t \leq \tau^{\epsilon,m,\rho}$,

$$\max(|b_t^{\Delta}|, |\sigma_t^{\Delta}|) \leq C(|\Delta_{(g,u),t}^{\epsilon,n}| + \rho).$$

By (Dembo & Zeitouni, 2010, Lemma 5.6.18), for all $\epsilon \in (0,1)$, $\delta > 0$,

$$\epsilon \sup_{\mu \in \mathcal{M}_1^{\epsilon}} \log \mathbb{P}_{(g,u) \sim \mu}\left( \sup_{t \in [0,\tau^{\epsilon,m,\rho}]} |\Delta_{(g,u),t}^{\epsilon,m}| \geq \delta \right) \leq K + \log\left(\frac{\rho^2}{\rho^2 + \delta^2}\right),$$

where $K$ is a constant independent of $\delta$, $\epsilon$, $\rho$, $\mu$ and $m$. Then

$$\lim_{\rho \to 0} \sup_{m \geq 1} \lim_{\epsilon \to 0} \epsilon \sup_{\mu \in \mathcal{M}_1^{\epsilon}} \log \mathbb{P}_{(g,u) \sim \mu}\left( \sup_{t \in [0,\tau^{\epsilon,m,\rho}]} |\Delta_{(g,u),t}^{\epsilon,m}| \geq \delta \right) = -\infty.$$

Since the event $\{\sup_{t \in [0,1]} |\Delta_{(g,u),t}^{\epsilon,m}| \geq \delta\}$ is contained in the union

$$\{\tau^{\epsilon,m,\rho} < 1\} \cup \{\sup_{t \in [0,\tau^{\epsilon,m,\rho}]} |\Delta_{(g,u),t}^{\epsilon,m}| \geq \delta\},$$

the lemma is proved if we show for all $\rho > 0$,

$$\lim_{m \to \infty} \lim_{\epsilon \to 0} \epsilon \sup_{\mu \in \mathcal{M}_1^{\epsilon}} \log \mathbb{P}_{(g,u) \sim \mu}\left( \sup_{t \in [0,1]} |Y_{(g,u),t}^{\epsilon,m} - Y_{(g,u),\frac{[mt]}{m}}^{\epsilon,m}| \geq \rho \right) = -\infty. \tag{23}$$

To prove this, recall that $|b|$ and $|\sigma|$ are bounded by a constant $H$, and

$$|Y_{(g,u),t}^{\epsilon,m} - Y_{(g,u),\frac{[mt]}{m}}^{\epsilon,m}| \leq H\left[\frac{1}{m} + \sqrt{\epsilon} \max_{k=0,\ldots,m-1} \sup_{0 \leq s \leq \frac{1}{m}} |u_{s+\frac{k}{m}} - u_{\frac{k}{m}}|\right].$$

Therefore, for $m > \frac{H}{\rho}$,

$$\sup_{\mu \in \mathcal{M}_1^{\epsilon}} \log \mathbb{P}_{(g,u) \sim \mu}\left( \sup_{0 \leq t \leq 1} |Y_{(g,u),t}^{\epsilon,m} - Y_{(g,u),\frac{[mt]}{m}}^{\epsilon,m}| \geq \rho \right)$$

$$\leq m\mathbb{P}\left( \sup_{0 \leq s \leq \frac{1}{m}} |u_s| \geq \frac{\rho - \frac{H}{m}}{\sqrt{\epsilon}C} \right)$$

$$\leq 4dme^{-m(\rho - \frac{H}{m})^2/2d\epsilon C^2}.$$

This guarantees (23) and proves the lemma. $\qquad \square$

*Proof of Theorem B.12.* First of all, notice that one can assume without loss of generality that $T = 1$ by rescaling the time interval $[0,T]$ to $[0,1]$. To see this, notice that the rescaled Brownian motion $B_{Tt}^d$ is equivalent to $\sqrt{T}B_t^d$ on $t \in [0,1]$ and if we make the change of variable $u_{Tt} = \sqrt{T}v_t$ accordingly, then

$$\int_0^T \frac{1}{2}|\dot{u}_t|^2 dt = \frac{1}{T}\int_0^1 \frac{1}{2}\left|\frac{d}{dt}u_{Tt}\right|^2 dt = \int_0^1 \frac{1}{2}|\dot{v}_t|^2 dt.$$

We will only deal with the $[0,1]$ interval below.

Define maps $F^m, F : C_H^{\text{Lip}}([0,1], D) \times C^0([0,1], \mathbb{R}^d) \to C^0([0,1], U)$ as follows.

For $F^m$, the image $f^m = F^m(g, u)$ satisfies $f_0^m = y_0$ and

$$f_t^m = f_{\frac{k}{m}}^m + \int_{\frac{k}{m}}^t b(f_{\frac{k}{m}}^m, g_s)\mathrm{d}s + \int_{\frac{k}{m}}^t \sigma(f_{\frac{k}{m}}^m, g_s)\dot{u}_s\mathrm{d}s, \tag{24}$$

on $t \in [\frac{k}{m}, \frac{k+1}{m}], k = 0, ..., m-1$.

For $F$, the image $f = F(g, u)$ instead satisfies $f_0 = y_0$ and

$$f_t = f_0 + \int_0^t b(f_s, g_s)\mathrm{d}s + \int_0^t \sigma(f_s, g_s)\dot{u}_s\mathrm{d}s \tag{25}$$

for $t \in [0,1]$.

It is not hard to check by Lipschitz boundedness of $b$, $\sigma$, and the compactnes of $D$ that $F^m$ and $F$ send $C_H^{\text{Lip}}([0,1], D) \times C^0([0,1], \mathbb{R}^d)$ to $C^0([0,1], U)$. Moreover, (21) and (22) can be reformulated as

$$Y_{(g,u),t}^{\epsilon,m} = F^m(g, u) \text{ and } Y_{(g,u),t}^{\epsilon} = F(g, u) \text{ respectively.} \tag{26}$$

For $(g, u), (g', u') \in C_H^{\text{Lip}}([0,1], D) \times C^0([0,1], \mathbb{R}^d)$, let

$$\Theta^m := |F^m(g, u) - F^m(g', u')|.$$

$\Theta^m$ is a function of $t$, and by the Lipschitz bounds on $b$ and $\sigma$,

$$\sup_{t \in [\frac{k}{m}, \frac{k+1}{m}]} \Theta^m(t) \le C(\Theta^m(\frac{k}{m}) + \|(g, u) - (g', u')\|_{C^0}).$$

Since $\Theta^m(0) = 0$, we derive the continuity of the operator $F^m$ by iterating the argument with $k = 0, 1, ..., m-1$. A similar argument guarantees the continuity of the operator $F$.

By (20) and Lemma B.16, for all family of initial values $\{z_0^\epsilon \in D\}_{\epsilon > 0}$,

$$\lim_{m \to \infty} \limsup_{\epsilon \to 0} \epsilon \log \mathbb{P}_{(g,u) \sim \mu_{(y_0, z_0^\epsilon), 1}^\epsilon}\Big( \sup_{t \in [0,1]} |Y_{(g,u),t}^{\epsilon,m} - Y_{(g,u),t}^{\epsilon}| \ge \delta \Big) = -\infty.$$

In other words, the families $\{\mu_{(y_0, z_0^\epsilon), 1}^\epsilon \circ (F^m)^{-1}\}$ are exponentially good approximations of the distribution of $Y_{(g,u),t}^\epsilon$ with $(g, u) \sim \mu_{(y_0, z_0^\epsilon), 1}^\epsilon$.

Thus if we can prove: for all $a < \infty$,

$$\lim_{m \to \infty} \sup_{\substack{(g,u) \in C_H^{\text{Lip}}([0,1], D) \times C^0([0,1], \mathbb{R}^d): \\ \int_0^1 \frac{1}{2}|\dot{u}_t|^2 \mathrm{d}t \le a}} \|F^m(g, u) - F(g, u)\|_{C^0} = 0, \tag{27}$$

then by Lemma B.7 and Lemma B.15, the distribution of $Y_{(g,u),t}^\epsilon$ with $(g, u) \sim \mu_{(y_0, z_0^\epsilon), 1}^\epsilon$ satisfies upper LDP with good rate function $y \mapsto \inf_{F(g,u)=y} I(g, u)$, which is exactly $\Phi_1(y)$. Because for each different value of $\epsilon$, $z_0^\epsilon$ is arbitrarily chosen in $D$ and the distribution of $Y_{(g,u),t}^\epsilon$ for $(g, u) \sim \mu_{(y_0, z_0^\epsilon), 1}^\epsilon$ coincides with the distribution $\nu_{(y_0, z_0^\epsilon), 1}^\epsilon$ of $Y_t^\epsilon$ with initial condition $(Y_0^\epsilon, Z_0^\epsilon) = (y_0, z_0^\epsilon)$ in (16), (17), this exactly yields the statement of the theorem.

It remains to show (27). Let

$$\Delta^m = |F^m(g, u) - F(g, u)|.$$

Note that if

$$\int_0^1 \frac{1}{2}|\dot{u}_t|^2 \mathrm{d}t \le a$$

holds, then for $f^m = F^m(g, u)$ and $f = F(g, u)$ given in (24), (25),

$$\sup_{t \in [0,1]} \max\left( |f_t^m - f_{\frac{[mt]}{m}}^m|, |f_t - f_{\frac{[mt]}{m}}| \right) \le C_0 \cdot \frac{1}{m} + \sqrt{C_0 a \cdot \frac{1}{m}}, \tag{28}$$

for some constant $C_0$ by the bound on the coefficients and Cauchy-Schwarz inequality. Write $\eta_m$ for the right hand side in (28). Then by Lipschitz continuity of $b$ and $\sigma$, for some other constant $C$,

$$
\begin{aligned}
\Delta_t^m =& |f_t^m - f_t| \\
=& \left| \int_0^t \left( b(f_{\frac{[ms]}{m}}^m, g_s) - b(f_s, g_s) \right) \mathrm{d}s + \int_0^t \left( \sigma(f_{\frac{[ms]}{m}}^m, g_s) - \sigma(f_s, g_s) \right) \dot{u}_s \mathrm{d}s \right| \\
\leq& \int_0^t C|f_{\frac{[ms]}{m}}^m - f_s| \mathrm{d}s + \int_0^t C|f_{\frac{[ms]}{m}}^m - f_s| \cdot |\dot{u}_s| \mathrm{d}s \\
\leq& \int_0^t C(\Delta_s^m + \eta_m)(1 + |\dot{u}_s|) \mathrm{d}s
\end{aligned}
$$

By Cauchy-Schwarz inequality, for all $t \in [0, 1]$

$$
\begin{aligned}
& (\Delta_t^m)^2 \\
\leq& C^2 \int_0^t (\Delta_s^m + \eta_m)^2 \mathrm{d}s \int_0^t (1 + |\dot{u}_s|)^2 \mathrm{d}s \\
\leq& C^2 \int_0^t 2\left((\Delta_s^m)^2 + (\eta_m)^2\right) \mathrm{d}s \int_0^t 2(1 + |\dot{u}_s|^2) \mathrm{d}s \\
\leq& 4C^2(1 + a)\left( \int_0^t (\Delta_s^m)^2 \mathrm{d}s + (\eta_m)^2 \right).
\end{aligned}
$$

Thus by Gronwall's inequality, $(\Delta_t^m)^2 \leq 4C^2(1 + a)e^{4C^2(1+a)t}(\eta_m)^2$. The equality (27) follows by letting $m \to \infty$, This completes the proof. $\qquad \square$

The following is a strengthen version of Theorem B.12.

**Theorem B.17.** *Given a closed subset $\mathcal{A}$ of the metric space $\mathcal{A} \subset C^0([0,1], U)$ and an initial value $(y_*, z_*) \in U \times V$, for solutions $(Y_t^\epsilon, Z_t^\epsilon)$ to (16) and (17), the following inequality holds:*

$$
\limsup_{\substack{\epsilon \to 0 \\ y_0 \to y_*}} \sup_{z_0 \in D} \epsilon \log \mathbb{P}_{(Y_0^\epsilon, Z_0^\epsilon) = (y_0, z_0)}(Y_t^\epsilon \in \mathcal{A}) \leq - \inf_{\substack{f \in \mathcal{A} \\ f_0 = y_*}} \Phi_T(f).
$$

*Proof of Theorem B.17.* As in the proof of Theorem B.12, we can assume $T = 1$.

By Theorem B.6, it suffices to prove that for any family of points $y_0^\epsilon \to y_*$ as $\epsilon \to 0$ and arbitrary $z_0^\epsilon \in D$, the family of distribution $\{\nu_{(y_0^\epsilon, z_0^\epsilon),1}^\epsilon\}$ is exponentially equivalent to $\{\nu_{(y_*, z_0^\epsilon),1}^\epsilon\}$ as in Definition B.5. Write

$$
\Delta_t^\epsilon = \left( \begin{array}{c} Y_{(y_0^\epsilon, z_0^\epsilon),t}^\epsilon - Y_{(y_*, z_0^\epsilon),t}^\epsilon \\ Z_{(y_0^\epsilon, z_0^\epsilon),t}^\epsilon - Z_{(y_*, z_0^\epsilon),t}^\epsilon \end{array} \right).
$$

Then $\Delta_t^\epsilon$ satisfies

$$
\mathrm{d}\Delta_t^\epsilon = \alpha_t^\epsilon \mathrm{d}t + \beta_t^\epsilon \mathrm{d}B_t^d
$$

where

$$
\alpha_t^\epsilon = \left( \begin{array}{c} b(Y_{(y_0^\epsilon, z_0^\epsilon),t}^\epsilon, Z_{(y_0^\epsilon, z_0^\epsilon),t}^\epsilon) - b(Y_{(y_*, z_0^\epsilon),t}^\epsilon, Z_{(y_*, z_0^\epsilon),t}^\epsilon) \\ h(Y_{(y_0^\epsilon, z_0^\epsilon),t}^\epsilon, Z_{(y_0^\epsilon, z_0^\epsilon),t}^\epsilon) - h(Y_{(y_*, z_0^\epsilon),t}^\epsilon, Z_{(y_*, z_0^\epsilon),t}^\epsilon) \end{array} \right),
$$

and

$$
\beta_t^\epsilon = \left( \begin{array}{c} \sigma(Y_{(y_0^\epsilon, z_0^\epsilon),t}^\epsilon, Z_{(y_0^\epsilon, z_0^\epsilon),t}^\epsilon) - \sigma(Y_{(y_*, z_0^\epsilon),t}^\epsilon, Z_{(y_*, z_0^\epsilon),t}^\epsilon) \\ 0 \end{array} \right).
$$

Moreover,

$$
\Delta_0^\epsilon = \left( \begin{array}{c} y_0^\epsilon - y_* \\ 0 \end{array} \right) \to 0 \text{ as } \epsilon \to 0.
$$

Remark that the coefficients $\alpha_t^\epsilon$, $\beta_t^\epsilon$ are progressively measurable processes with respect to the filter generated by the Brownian motion $\{B_t^d\}$. Moreover, by Lipschitz continuity of $b$, $\sigma$, $h$, for some constant $C_0$,

$$
\max(|\alpha_t^\epsilon|, |\beta_t^\epsilon|) \leq C_0 |\Delta_t^\epsilon|.
$$

By applying (Dembo & Zeitouni, 2010, Lemma 5.6.18) with $\tau_1 = 1$, we know that there is another constant $C$ such that for all $\rho > 0$, $\delta > 0$,

$$\epsilon \log \mathbb{P}(\sup_{t \in [0,1]} |\Delta_t^\epsilon| \geq \delta) \leq C + \log \frac{\rho^2 + |y_0^\epsilon - y_*|^2}{\rho^2 + \delta^2}.$$

By letting $\rho \to 0$ first and then $\epsilon \to 0$, it follows that

$$\limsup_{\epsilon \to 0} \epsilon \log \mathbb{P}(\sup_{t \in [0,1]} |\Delta_t^\epsilon| \geq \delta) = -\infty.$$

This shows $\{\nu_{(y_0^\epsilon, z_0^\epsilon), 1}^\epsilon\}$ is exponentially equivalent to $\{\nu_{(y_*, z_0^\epsilon), 1}^\epsilon\}$, which suffices to conclude the proof. $\qquad\square$

**Corollary B.18.** *Given a closed subset $\mathcal{A}$ of the metric space $\mathcal{A} \subset C^0([0,1], U)$ and a compact set $K \subseteq U$, for solutions $(Y_t^\epsilon, Z_t^\epsilon)$ to (16) and (17), the following inequality holds:*

$$\limsup_{\epsilon \to 0} \epsilon \log \sup_{(y_0, z_0) \in K \times D} \mathbb{P}_{(Y_0^\epsilon, Z_0^\epsilon) = (y_0, z_0)}(Y_t^\epsilon \in \mathcal{A}) \leq - \inf_{\substack{f \in \mathcal{A} \\ f_0 \in K}} \Phi_T(f).$$

*Proof.* Let $M \in [0, \infty)$ be a finite value strictly less than $\inf_{\substack{f \in \mathcal{A} \\ f_0 \in K}} \Phi_T(f) \in [0, \infty]$. By Theorem B.17, for all $y \in K$, there is a value $\epsilon_y$ such that for all $y_0 \in B_{\epsilon_y}(y)$ and $0 < \epsilon < \epsilon_y$,

$$\epsilon \sup_{z_0 \in D} \epsilon \log \mathbb{P}_{(Y_0^\epsilon, Z_0^\epsilon) = (y_0, z_0)}(Y_t^\epsilon \in \mathcal{A}) \leq -M.$$

Cover $K$ by finitely many balls of the form $B_{\epsilon_{y_i}}(y_i)$, then for all $0 < \epsilon < \min_i \epsilon_{y_i}$,

$$\epsilon \sup_{y \in K} \sup_{z_0 \in D} \epsilon \log \mathbb{P}_{(Y_0^\epsilon, Z_0^\epsilon) = (y_0, z_0)}(Y_t^\epsilon \in \mathcal{A}) \leq -M.$$

The proof is completed by letting $M \to \inf_{\substack{f \in \mathcal{A} \\ f_0 \in K}} \Phi_T(f)$. $\qquad\square$

### B.3 Exiting time from basin

In this section, it will be assumed, in addition to Assumption B.8, that:

**Assumption B.19.** *There are a $C^2$ function $L : U \to [0, \infty)$ and a bounded open set $V \subset U$ such that:*

*(1) $\nabla_{b(y,z)} L(y) \leq 0$ for all $(y, z) \in V \times D$ and the equality holds if and only if $L(y) = 0$;*

*(2) $L$ is strictly positive on $\partial V$.*

Write $V_q = L^{-1}([0, q)) \cap V$ and choose $q_0$ sufficiently small such that $\overline{V_{q_0}}$ lies in the interior of $V$. In particular, for all $q \in [0, q_0]$, $L|_{\partial V_q} \equiv q$.

**Definition B.20.** *Suppose $0 < q < Q \leq q_0$. For a solution $(Y_t^\epsilon, Z_t^\epsilon)$ of (16), (17) with initial value in $V_Q \times D$, denote by $\tau_{q,Q}^\epsilon$ the first time $Y_t^\epsilon$ hits $\overline{V_q} \cup \partial V_Q$, and by $\tau_Q^\epsilon$ the first time $Y_t^\epsilon$ hits $\partial V_Q$.*

Our goal is to prove the following main theorem:

**Theorem B.21.** *Under Assumptions B.8 and B.19, for all $0 < q < Q < q_0$ there exists $\mathcal{I}_Q > 0$ such that for all,*

$$\lim_{\epsilon \to 0} \sup_{(y_0, z_0) \in V_q \times D} \mathbb{P}_{(Y_0^\epsilon, Z_0^\epsilon) = (y_0, z_0)}(\tau_Q^\epsilon > e^{\frac{\mathcal{I}_Q}{\epsilon}}) = 1.$$

The following lemma is an analogue to (Dembo & Zeitouni, 2010, Lemma 5.7.22)

**Lemma B.22.** *For all $0 < q < Q' < Q \leq q_0$, the stopping time $\tau_{q,Q}^\epsilon$ satisfies*

$$\lim_{\epsilon \to 0} \sup_{(y_0, z_0) \in V_{Q'} \times D} \mathbb{P}_{(Y_0^\epsilon, Z_0^\epsilon) = (y_0, z_0)}(\tau_{q,Q}^\epsilon < \infty \text{ and } Y_{\tau_{q,Q}^\epsilon}^\epsilon \in \overline{V_q}) = 1.$$

*Proof.* Consider the solution $Y_t^0$ to the deterministic flow

$$\mathrm{d}Y_t^0 = b(Y_t^0, z_0)\mathrm{d}t \tag{29}$$

starting at $y_0$. By Assumption B.19, $L(Y_t^0)$ is decreasing and must converge to 0 as $t \to 0$, and $Y_t^0$ remains in $V_Q$. Denote by $\tilde{T}$ the first moment $L(Y_t^0)$ reaches $\frac{q}{2}$. Set

$$\delta = \|L\|_{C^1}^{-1} \min(\frac{q}{4}, \frac{Q - Q'}{2}) > 0.$$

Notice $\sup_{t \in [0, \tilde{T}]} \mathrm{dist}\big((Y_t^\epsilon, Z_t^\epsilon), (Y_t^0, z_0)\big) < \delta$ implies

$$\sup_{t \in [0, \tilde{T}]} |L(Y_t^\epsilon) - L(Y_t^0)| \le \min(\frac{1}{4}q, \frac{Q - L(y_0)}{2}).$$

Since $L(Y_t^0) \le L(Y_0) = Q'$ and $L(Y_{\tilde{T}}^0) = \frac{q}{2}$, this guarantees that

$$L(Y_t^\epsilon) \le L(Y_0) + \|L\|_{C^1}\delta < Q$$

and

$$L(Y_{\tilde{T}}^\epsilon) \le \frac{q}{2} + \|L\|_{C^1}\delta < q.$$

It would in turn follow that $\tau_{q,Q}^\epsilon \le \tilde{T} < \infty$ and $Y_{\tau_{q,Q}^\epsilon}^\epsilon \in V_q$. Hence it suffices to prove

$$\lim_{\epsilon \to 0} \mathbb{P}_{(Y_0^\epsilon, Z_0^\epsilon) = (y_0, z_0)} \Big( \sup_{t \in [0, \tilde{T}]} \big|(Y_t^\epsilon, Z_t^\epsilon), (Y_t^0, z_0)\big| < \delta \Big) = 1. \tag{30}$$

For simplicity, write $J_t^\epsilon := \big|(Y_t^\epsilon, Z_t^\epsilon), (Y_t^0, z_0)\big|$. Let $M < \infty$ be a common upper bound on the $C^0$ and the Lipschitz norms of the maps $b$, $\sigma$ and $h$ over the domains $\epsilon \in [0, 1]$, $y \in V_{q_0}$, $z \in D$. Then $J_0^\epsilon = 0$ and

$$J_t^\epsilon \le \int_0^t 2M J_s^\epsilon \mathrm{d}s + \sqrt{\epsilon} \Big| \int_0^t \sigma(Y_s^\epsilon, Z_s^\epsilon)\mathrm{d}B_s^d \Big|.$$

By Gronwall's inequality,

$$\sup_{t \in [0, \tilde{T}]} J_t^\epsilon \le \sqrt{\epsilon} e^{2M\tilde{T}} \sup_{t \in [0, \tilde{T}]} \Big| \int_0^t \sigma(Y_s^\epsilon, Z_s^\epsilon)\mathrm{d}B_s^d \Big|.$$

Therefore, in order to make $\sup_{t \in [0, \tilde{T}]} J_t^\epsilon \ge \delta$, we must have

$$\sup_{t \in [0, \tilde{T}]} \Big| \int_0^t \sigma(Y_s^\epsilon, Z_s^\epsilon)\mathrm{d}B_s^d \Big| \ge \delta \epsilon^{-\frac{1}{2}} e^{-2M\tilde{T}},$$

and thus

$$\sup_{t \in [0, \tilde{T}]} \Big| \int_0^t \sigma_k(Y_s^\epsilon, Z_s^\epsilon)\mathrm{d}B_s^1 \Big| \ge \frac{\delta}{d} \epsilon^{-\frac{1}{2}} e^{-2M\tilde{T}}$$

must hold for at least one of the row vectors $\sigma_k$ of $\sigma$.

The process $G_{k,t}^\epsilon = \int_0^t \sigma_k(Y_s^\epsilon, Z_s^\epsilon)\mathrm{d}B_s^1$ is a continuous martingale with increasing process $\langle G_k^\epsilon \rangle_t \le M^2 t$ almost surely.

Recall $\langle G_k^\epsilon \rangle_t$ is defined as $\int_0^t \sigma_k^2(Y_s^\epsilon, Z_s^\epsilon)\mathrm{d}s$. By the Burkholder-Davis-Gundy inequality (see e.g. (Dembo & Zeitouni, 2010, Chapter E)),

$$\mathbb{E}_{(Y_0^\epsilon, Z_0^\epsilon) = (y_0, z_0)} \Big( \sup_{t \in [0, \tilde{T}]} |G_{k,t}^\epsilon|^2 \Big) \le C \mathbb{E}_{(Y_0^\epsilon, Z_0^\epsilon) = (y_0, z_0)} \langle G_k^\epsilon \rangle_{\tilde{T}} \le C M^2 \tilde{T}$$

for an absolute constant $C$. And thus by the Chebyshev's theorem

$$\mathbb{P}_{(Y_0^\epsilon, Z_0^\epsilon) = (y_0, z_0)} \Big( \sup_{t \in [0, \tilde{T}]} \Big| \int_0^t \sigma_k(Y_s^\epsilon, Z_s^\epsilon)\mathrm{d}B_s^1 \Big| \ge \frac{\delta}{d} \epsilon^{-\frac{1}{2}} e^{-2M\tilde{T}} \Big)$$

$$\le C M^2 \tilde{T} (\frac{\delta}{d} \epsilon^{-\frac{1}{2}} e^{-2M\tilde{T}})^{-2}.$$

From the earlier discussion, after summing over all $k$'s, we obtain that

$$\mathbb{P}_{(Y_0^\epsilon, Z_0^\epsilon)=(y_0,z_0)}\big(\sup_{t\in[0,\tilde{T}]} J_t^\epsilon \geq \delta\big) \leq CM^2\tilde{T}(\frac{\delta}{d}\epsilon^{-\frac{1}{2}}e^{-2M\tilde{T}})^{-2}. \tag{31}$$

We deduce (30) from (31) by letting $\epsilon \to 0$, which concludes the proof. $\qquad\square$

The following lemma is the analogue of (Dembo & Zeitouni, 2010, Lemma 5.7.23).

**Lemma B.23.** *For all $\delta, a > 0$ and bounded set $K \subset U$, there exists a constant $T_0 = T_0(\delta, a, K) > 0$ such that*

$$\limsup_{\epsilon\to 0} \epsilon \log \sup_{(y_0,z_0)\in K\times D} \mathbb{P}_{(Y_0^\epsilon, Z_0^\epsilon)=(y_0,z_0)}\big(\sup_{t\in[0,T_0]} |(Y_t^\epsilon - y_0, Z_t^\epsilon - z_0)| \geq \delta\big) < -a.$$

*Proof.* Without loss of generality, assume $\epsilon, \delta \in [0,1]$ with $\delta$ fixed and $\epsilon$ varying. Let the stopping time $\zeta^\epsilon$ be the first time such that $|(Y_t^\epsilon - y_0, Z_t^\epsilon - z_0)| \geq \delta$. Then for every $t \in [0, \zeta^\epsilon]$,

$$|b(Y_t^\epsilon, Z_t^\epsilon)| \leq \max_{B_\delta(K)\times D} |b| + \|b\|_{\mathrm{Lip}}\delta,$$

$$|\sigma(Y_t^\epsilon, Z_t^\epsilon)| \leq \max_{B_\delta(K)\times D} |\sigma| + \|\sigma\|_{\mathrm{Lip}}\delta,$$

$$|h(\epsilon, Y_t^\epsilon, Z_t^\epsilon)| \leq \max_{[0,1]\times B_\delta(K)\times D} |h| + \|h\|_{\mathrm{Lip}}\delta,$$

and thus they all are uniformly bounded by a constant $M$. For all $0 \leq t \leq \min(\zeta^\epsilon, \frac{\delta}{4M})$, $|Z_t^\epsilon - z_0| \leq \frac{\delta}{4}$, and $|Y_t^\epsilon - y_0| \leq \frac{\delta}{4} + |\int_0^t \sqrt{\epsilon}\sigma(Y_s^\epsilon, Z_s^\epsilon)\mathrm{d}B_s^d|$. Hence for any $T_0 \leq \frac{\delta}{4M}$ we have

$$\mathbb{P}_{(Y_0^\epsilon, Z_0^\epsilon)=(y_0,z_0)}(\zeta^\epsilon \geq T_0)$$

$$\leq \mathbb{P}_{(Y_0^\epsilon, Z_0^\epsilon)=(y_0,z_0)}\Big(\zeta^\epsilon \geq T_0 \text{ and } \sqrt{\epsilon}\sup_{t\in[0,T]} \Big|\int_0^t \sigma(Y_s^\epsilon, Z_s^\epsilon)\mathrm{d}B_s^d\Big| \geq \frac{\delta}{2}\Big).$$

As in the proof of (Dembo & Zeitouni, 2010, Lemma 5.7.23), it suffices to consider each row vector $\sigma_k$ of $\sigma$. The stochastic process $\int_0^t \sigma_k(Y_s^\epsilon, Z_s^\epsilon)\mathrm{d}B_s^1$ is equivalent to $B_{\tau_{k,t}^\epsilon}^1$ by the time change theorem (see (Dembo & Zeitouni, 2010, Chapter E.2)) where $\tau_{k,t}^\epsilon = \int_0^t \sigma_k^2(Y_s^\epsilon, Z_s^\epsilon)\mathrm{d}s$. The function $\tau_{k,t}^\epsilon$ is increasing in $t$ and almost surely $\tau_{k,t}^\epsilon \leq M^2 t$ if $T_0 \leq \zeta^\epsilon$ as the $C^0$ and the Lipschitz norms of $b, \sigma, h$ are bounded by $M$ before the stopping time $\zeta^\epsilon$. By (Dembo & Zeitouni, 2010, Lemma 5.2.1),

$$\mathbb{P}_{(Y_0^\epsilon, Z_0^\epsilon)=(y_0,z_0)}\Big(\zeta^\epsilon \geq T_0 \text{ and } \sqrt{\epsilon}\sup_{t\in[0,T_0]} \Big|\int_0^t \sigma_k(Y_s^\epsilon, Z_s^\epsilon)\mathrm{d}B_s^1\Big| \geq \frac{\delta}{2d}\Big)$$

$$\leq \mathbb{P}_{(Y_0^\epsilon, Z_0^\epsilon)=(y_0,z_0)}\Big(\zeta^\epsilon \geq T_0 \text{ and } \sqrt{\epsilon}\sup_{t\in[0,T_0]} B_{\tau_{k,t}^\epsilon}^1 \geq \frac{\delta}{2d}\Big)$$

$$\leq \mathbb{P}_{(Y_0^\epsilon, Z_0^\epsilon)=(y_0,z_0)}\Big(\zeta^\epsilon \geq T_0 \text{ and } \sqrt{\epsilon}\sup_{\tau\in[0,M^2 T_0]} B_\tau^1 \geq \frac{\delta}{2d}\Big)$$

$$\leq \mathbb{P}_{(Y_0^\epsilon, Z_0^\epsilon)=(y_0,z_0)}\Big(\sqrt{\epsilon}\sup_{\tau\in[0,M^2 T_0]} B_\tau^1 \geq \frac{\delta}{2d}\Big)$$

$$\leq 4e^{-(\frac{\delta}{2d})^2/(2\epsilon M^2 T_0)}.$$

Summing over different $k$'s, we conclude from the two inequalities above that for all $T_0 \in [0, \frac{\delta}{4M}]$

$$\mathbb{P}_{(Y_0^\epsilon, Z_0^\epsilon)=(y_0,z_0)}(\zeta^\epsilon \geq T_0) \leq d \cdot 4e^{-(\frac{\delta}{2d})^2/(2\epsilon M^2 T_0)} = 4de^{-(\frac{1}{8}\delta^2 d^{-2}M^{-2}T_0^{-1})/\epsilon}.$$

It now suffices to take $T_0 = \min(\frac{\delta}{4M}, \frac{\delta^2}{8d^2 M^2 a})$. $\qquad\square$

**Definition B.24.** *For $y_0, y_1 \in U$ and $T > 0$, define an energy cost $\Psi_T(y_0, y_1)$ by*

$$\Psi_T(y_0, y_1) = \inf\Big\{\int_0^T \frac{1}{2}|\dot{u}_s|^2\mathrm{d}s : (f, g, u) \in \mathcal{S}_T, f_0 = y_0, \ f_T = y_1\Big\}$$

$$= \inf\big\{\Phi(f) : f \in C^0([0,T], U)\big\}.$$

The following lemma claims there is a minimum energy cost required to increase the loss function $L$ between different levels.

**Lemma B.25.** *For all* $0 < q < Q \leq q_0$,

$$\inf_{T \geq 0} \inf_{(y_0, y_1) \in \overline{V_q} \times \partial V_Q} \Psi_T(y_0, y_1) > 0. \tag{32}$$

*Moreover, there is constant* $T_{q,Q} > 0$ *such that*

$$\inf_{T \geq T_{q,Q}} \inf_{y_0, y_1 \in \overline{V_Q} \backslash V_q} \Psi_T(y_0, y_1) > 0. \tag{33}$$

*Proof.* Suppose for now $y_0, y_1 \in \overline{V_Q} \backslash \partial V_q$, and $(f, g, u) \in \mathcal{S}_T$ with $f_0 = y_0$, $f_T = y_1$. Then by (18),

$$\begin{aligned}
\mathrm{d}L(f_t) &= \nabla^\top L(f_t) \mathrm{d}f_t \\
&= \big(\nabla^\top L(f_s)\big) b(f_s, g_s) \mathrm{d}s + \big(\nabla^\top L(f_s)\big) \sigma(f_s, g_s) \mathrm{d}u_s.
\end{aligned}$$

Because $q > 0$, by Assumption B.19, as $\big(\nabla^\top L(y)\big) b(y, z) = \nabla_{b(y,z)} L(y) < 0$ for all $y \in \overline{V_Q} \backslash V_q$ and $z \in D$. In particular, this also shows $\nabla L(y) \neq 0$ for all $y \in \overline{V_Q} \backslash V_q$. Since both $\overline{V_Q} \backslash V_q$ and $D$ are compact, There exists positive constant $\kappa = \kappa(q, Q) > 0$ and $\eta = \eta(q, Q) > 0$, such that $\big(\nabla^\top L(y)\big) b(y, z) \leq -\kappa |\nabla L(y)|^2$ and $|\nabla L(y)| \geq \eta$ for $(y, z) \in (\overline{V_Q} \backslash V_q) \times D$.

Integrating from $0$ to $T$, we get

$$\begin{aligned}
&L(y_1) - L(y_0) \\
=&L(f_T) - L(f_0) = \int_0^T \mathrm{d}L(f_t) \\
=&\int_0^T \left( \big(\nabla^\top L(f_s)\big) b(f_s, g_s) \mathrm{d}s + \big(\nabla^\top L(f_s)\big) \sigma(f_s, g_s) \mathrm{d}u_s \right) \\
\leq& -\kappa \int_0^T |\nabla L(f_s)|^2 \mathrm{d}s \\
&+ \left( \int_0^T |\nabla L(f_s)|^2 \mathrm{d}s \right)^{\frac{1}{2}} \left( \int_0^T |\sigma(f_s, g_s)\dot{u}_s|^2 \mathrm{d}s \right)^{\frac{1}{2}} \\
\leq& -\kappa \int_0^T |\nabla L(f_s)|^2 \mathrm{d}s \\
&+ \left( \int_0^T |\nabla L(f_s)|^2 \mathrm{d}s \right)^{\frac{1}{2}} \|\sigma\|_{C^0(\overline{V_Q} \times D)} \left( \int_0^T |\dot{u}_s|^2 \mathrm{d}s \right)^{\frac{1}{2}}.
\end{aligned} \tag{34}$$

In order to show (32), note that if $y_0 \in \overline{V_q}$ and $y_1 \in \partial V_Q$, then $L(y_1) - L(y_0) = Q - q > 0$. By (34) and Cauchy-Schwarz inequality,

$$\begin{aligned}
Q - q \leq& -\kappa \int_0^T |\nabla L(f_s)|^2 \mathrm{d}s \\
&+ \left( \int_0^T |\nabla L(f_s)|^2 \mathrm{d}s \right)^{\frac{1}{2}} \|\sigma\|_{C^0(\overline{V_Q} \times D)} \left( \int_0^T |\dot{u}_s|^2 \mathrm{d}s \right)^{\frac{1}{2}} \\
\leq& \frac{1}{4\kappa} \|\sigma\|_{C^0(\overline{V_Q} \times D)}^2 \int_0^T |\dot{u}_s|^2 \mathrm{d}s.
\end{aligned} \tag{35}$$

Since $\sigma$ is continuous, $\|\sigma\|_{C^0(\overline{V_Q} \times D)} < \infty$. It follow that

$$\int_0^T \frac{1}{2} |\dot{u}_s|^2 \mathrm{d}s \geq 2\kappa(Q - q) \|\sigma\|_{C^0(\overline{V_Q} \times D)}^{-2} > 0.$$

This provides a positive lower bound for $\Psi_T(y_0, y_1)$ that is uniform for $T > 0$, $y_0 \in \overline{V_q}$ and $y_1 \in \partial V_Q$. This proves (32).

We now prove (33). Note that for all $y_0, y_1 \in \overline{V_Q}\backslash V_q$, $L(y_1) - L(y_0) \geq q - Q$. By (34),

$$\left(\int_0^T |u_s|^2 \mathrm{d}s\right)^{\frac{1}{2}}$$

$$\geq \|\sigma\|_{C^0(\overline{V_Q}\times D)}^{-1}\left(\kappa\left(\int_0^T |\nabla L(f_s)|^2 \mathrm{d}s\right)^{\frac{1}{2}} - (Q-q)\left(\int_0^T |\nabla L(f_s)|^2 \mathrm{d}s\right)^{-\frac{1}{2}}\right)$$

$$\geq \|\sigma\|_{C^0(\overline{V_Q}\times D)}^{-1}(\kappa\eta T^{\frac{1}{2}} - (Q-q)\eta^{-1}T^{-\frac{1}{2}}).$$

The last expression is uniformly positive for $T \geq T_{q,Q} := 2(Q-q)\kappa^{-1}\eta^{-2}$. Thus $\Psi_T(y_0, y_1)$ is uniformly positive over the region given by $T \geq T_{q,Q}$, $y_0, y_1 \in \overline{V_Q}\backslash\partial V_q$. This proves (33). □

**Lemma B.26.** *For all $0 < q < Q \leq q_0$ and initial value $(y_0, z_0) \in V_Q \times D$, the stopping time $\tau_{q,Q}^\epsilon$ satisfies*

$$\lim_{T\to\infty}\limsup_{\epsilon\to 0}\epsilon\log\sup_{(y_0,z_0)\in V_Q\times D}\mathbb{P}_{(Y_0^\epsilon, Z_0^\epsilon)=(y_0,z_0)}(\tau_{q,Q}^\epsilon > T) = -\infty.$$

*Proof.* Note that $\tau_{q,Q}^\epsilon > T$ implies $Y_t^\epsilon \in V_Q\backslash\overline{V_q} \subseteq \overline{V_Q}\backslash V_q$ for all $t \in [0, T]$; or in other words, $\{Y_t^\epsilon\}_{t\in[0,T]} \in C^0([0,T], \overline{V_Q}\backslash V_q)$.

Therefore, by Corollary B.18,

$$\lim_{T\to\infty}\limsup_{\epsilon\to 0}\epsilon\log\sup_{(y_0,z_0)\in V_Q\times D}\mathbb{P}_{(Y_0^\epsilon, Z_0^\epsilon)=(y_0,z_0)}(\tau_{q,Q}^\epsilon > T)$$

$$\leq -\inf_{f\in C^0([0,T],\overline{V_Q}\backslash V_q)}\Phi(f);$$

and in consequence it suffices to show

$$\lim_{T\to\infty}\inf_{f\in C^0([0,T],\overline{V_Q}\backslash V_q)}\Phi(f) = \infty. \tag{36}$$

Assume, for the sake of contradiction, that (36) is false, then for some $M < \infty$ and any all $k \in \mathbb{N}$, there exists $f_k \in C^0([0, kT_{q,Q}], \overline{V_Q}\backslash V_q)$ with $\Phi(f_k) \leq M$, where $T_{q,Q}$ is given by Lemma B.25. After breaking $f_k$ into $k$ segments on subintervals of length $T_{q,Q}$, it follows that there exists $f_k^* \in C^0([0, T_{q,Q}], \overline{V_Q}\backslash V_q)$ such that $\Phi(f_k^*) \leq \frac{M}{k}$.

By taking limit in $C^0$ norm (which is permitted by Lemma B.10) and using the lower semicontinuity of the good rate function $\Phi$, there exists $f^* \in C^0([0, T_{q,Q}], \overline{V_Q}\backslash V_q)$ with $\Phi(f^*) = 0$. This implies $\Psi_T(y_0, y_1) = 0$ and thus it contradicts to the inequality (33) of Lemma B.25. This completes the proof. □

Denote by $\mathcal{I}_{q,Q} > 0$ the left hand side in (32).

**Lemma B.27.** *The solutions to (16) and (17) satisfy*

$$\lim_{q\to 0}\limsup_{\epsilon\to 0}\epsilon\log\sup_{(y_0,z_0)\in V_{2q}\times D}\mathbb{P}_{(Y_0^\epsilon, Z_0^\epsilon)=(y_0,z_0)}(Y_{\tau_{q,Q}^\epsilon}^\epsilon \in \partial V_Q) \leq -\lim_{q\to 0}\mathcal{I}_{q,Q}. \tag{37}$$

*Proof.* Fix an arbitrarily small $\delta$, and let $\widehat{\mathcal{I}}_{Q,\delta} := \min(\lim_{q\to 0}\mathcal{I}_{q,Q} - \delta, \frac{1}{\delta})$. Note that the right hand side in (37) always exists because $\mathcal{I}_{q,Q}$ is a decreasing function in $q$ by construction. In particular, $\mathcal{I}_{2q,Q} \geq \widehat{\mathcal{I}}_{Q,\delta}$ when $q$ is sufficiently small depending on $Q$ and $\delta$.

By Lemma B.26, there exists a large $T_* = T_*(q, Q, \delta) < \infty$ such that

$$\limsup_{\epsilon\to 0}\epsilon\log\sup_{(y_0,z_0)\in V_{2q}\times D}\mathbb{P}_{(Y_0^\epsilon, Z_0^\epsilon)=(y_0,z_0)}(\tau_{q,Q}^\epsilon > T_*) \leq -\widehat{\mathcal{I}}_{Q,\delta}. \tag{38}$$

In addition,

$$\inf_{\substack{f\in C^0([0,T_*],U):\\ f_0\in V_{2q},\ \sup_{t\in[0,T_*]}L(f_t)\geq Q}}\Phi(f) \geq \mathcal{I}_{2q,Q} \geq \widehat{\mathcal{I}}_{Q,\delta},$$

and thus by Corollary B.18,

$$\limsup_{\epsilon \to 0} \epsilon \log \sup_{(y_0,z_0) \in V_{2q} \times D} \mathbb{P}_{(Y_0^\epsilon, Z_0^\epsilon)=(y_0,z_0)} \big( \sup_{t \in [0,T_*]} L(Y_t^\epsilon) \geq Q \big) \leq -\widehat{\mathcal{I}}_{Q,\delta}. \tag{39}$$

Note that the event $\{Y_{\tau_{q,Q}^\epsilon}^\epsilon \in \partial V_Q\}$ is contained in the union of the events $\{\tau_{q,Q}^\epsilon > T_*\}$ and $\{\sup_{t \in [0,T_*]} L(Y_t^\epsilon) \geq Q\}$. Therefore, we obtain by combining (38) and (39) that the inequality

$$\limsup_{\epsilon \to 0} \epsilon \log \sup_{(y_0,z_0) \in V_{2q} \times D} \mathbb{P}_{(Y_0^\epsilon, Z_0^\epsilon)=(y_0,z_0)} (Y_{\tau_{q,Q}^\epsilon}^\epsilon \in \partial V_Q) \leq -\widehat{\mathcal{I}}_{Q,\delta} \tag{40}$$

holds for all sufficiently small $q$. The lemma then follows by letting $\delta \to 0$. $\qquad \square$

We are now ready to establish Theorem B.21.

*Proof of Theorem B.21.* Choose $q < \frac{Q}{2}$ to be sufficiently small. Define a sequence of stopping times $\theta_0 < \tau_0 < \theta_1 < \tau_1 < \cdots$ recursively by letting

$$\theta_0 := 0,$$
$$\tau_m := \inf\{t > \theta_m : Y_t^\epsilon \in \overline{V_q} \cup \partial V_Q\},$$
$$\theta_m := \inf\{t > \tau_{m-1} : Y_t^\epsilon \in \partial V_{2q}\}.$$

By Lemma B.22, all these stopping times are finite almost surely. Write $\widetilde{Y}_m = Y_{\tau_m}^\epsilon$, which is a Markov chain.

Recall that by Lemma B.25 $\mathcal{I}_{q,Q} > 0$ for all $0 < q < Q$ and is an decreasing function in $q$. Let $\mathcal{I}_Q := \frac{1}{2} \lim_{q \to 0} \mathcal{I}_{q,Q} > 0$ and fix $0 < \alpha < \frac{1}{4}\mathcal{I}_Q$. By Lemma B.27, if we fix a sufficiently small $q$, then

$$\limsup_{\epsilon \to 0} \epsilon \log \sup_{(y_0,z_0) \in V_{2q} \times D} \mathbb{P}_{(Y_0^\epsilon, Z_0^\epsilon)=(y,z)} (Y_{\tau_{q,Q}^\epsilon}^\epsilon \in \partial V_Q) \leq -\mathcal{I}_Q - 3\alpha.$$

By plugging in $(Y_{\theta_m}^\epsilon, Z_{\theta_m}^\epsilon)$ as the the value for $(y,z)$, we deduce that there exists $\epsilon_0 > 0$ such that for all $0 < \epsilon < \epsilon_0$ and $m \geq 1$,

$$\sup_{(y_0,z_0) \in \overline{V_Q} \times D} \epsilon \log \mathbb{P}_{(Y_0^\epsilon, Z_0^\epsilon)=(y_0,z_0)} (\tau_Q^\epsilon = \tau_m) \leq -\mathcal{I}_Q - 2\alpha. \tag{41}$$

On the other hand, assuming $\epsilon_0$ is sufficiently small, applying Lemma B.23 with $a = \mathcal{I}_Q + 2\alpha$, $\delta = \frac{1}{2}q$ and $K = \overline{V_Q}$ yields that, for some fixed $T_0 > 0$ depending on $q$ and $Q$ and all $\epsilon \in (0, \epsilon_0]$.

$$\sup_{(y_0,z_0) \in \overline{V_Q} \times D} \epsilon \log \mathbb{P}_{(Y_0^\epsilon, Z_0^\epsilon)=(y_0,z_0)} (\theta_m - \tau_{m-1} \leq T_0)$$
$$\leq \sup_{(y,z) \in \overline{V_Q} \times D} \epsilon \log \mathbb{P}_{(Y_0^\epsilon, Z_0^\epsilon)=(y,z)} \big( \sup_{t \in [0,T_0]} |(Y_t^\epsilon - y, Z_t^\epsilon - z)| \geq \frac{1}{2}q \big) \tag{42}$$
$$< -\mathcal{I}_Q - 2\alpha.$$

For a given $M$, the event $\{\tau_Q^\epsilon \leq MT_0\}$ is contained in the union of the events $\bigcup_{m=0}^M \{\tau_Q^\epsilon = \tau_m\}$ and $\bigcup_{m=1}^M \{\theta_m - \tau_{m-1} \leq T_0\}$. Combining this fact and the inequalities (41), (42) yield

$$\sup_{(y_0,z_0) \in V_q \times D} \mathbb{P}_{(Y_0^\epsilon, Z_0^\epsilon)=(y_0,z_0)} (\tau_Q^\epsilon \leq MT_0)$$
$$\leq \sup_{(y_0,z_0) \in V_q \times D} \mathbb{P}_{(Y_0^\epsilon, Z_0^\epsilon)=(y_0,z_0)} (\tau_Q^\epsilon = \tau_0) + \sum_{m=1}^M \sup_{(y_0,z_0) \in V_q \times D} \mathbb{P}_{(Y_0^\epsilon, Z_0^\epsilon)=(y_0,z_0)} (\tau_Q^\epsilon = \tau_m)$$
$$+ \sum_{m=1}^M \sup_{(y_0,z_0) \in V_q \times D} \mathbb{P}_{(Y_0^\epsilon, Z_0^\epsilon)=(y_0,z_0)} (\theta_m - \tau_{m-1} \leq T_0) \tag{43}$$
$$\leq \sup_{(y_0,z_0) \in V_q \times D} \mathbb{P}_{(Y_0^\epsilon, Z_0^\epsilon)=(y_0,z_0)} (\tau_Q^\epsilon = \tau_0) + 2Me^{-\frac{\mathcal{I}_Q + 2\alpha}{\epsilon}}.$$

Set $M = \lceil T_0^{-1} e^{\frac{\mathcal{I}_Q + \alpha}{\epsilon}} \rceil$, which makes $MT_0 \geq e^{\frac{\mathcal{I}_Q}{\epsilon}}$ for all sufficiently small $\epsilon$. Moreover as $\epsilon \to 0$, $\sup_{(y_0, z_0) \in V_q \times D} \mathbb{P}_{(Y_0^\epsilon, Z_0^\epsilon) = (y_0, z_0)} (\tau_Q^\epsilon = \tau_0) \to 0$ by Lemma B.22; and $2M e^{-\frac{\mathcal{I}_Q + 2\alpha}{\epsilon}} \to 0$ by the choice of $M$. Thus

$$\sup_{(y_0, z_0) \in V_q \times D} \mathbb{P}_{(Y_0^\epsilon, Z_0^\epsilon) = (y_0, z_0)} (\tau_Q^\epsilon \leq e^{\frac{\mathcal{I}_Q}{\epsilon}}) \leq \sup_{(y_0, z_0) \in V_q \times D} \mathbb{P}_{(Y_0^\epsilon, Z_0^\epsilon) = (y_0, z_0)} (\tau_Q^\epsilon \leq MT_0)$$

which tends to 0 as $\epsilon \to 0$. This completes the proof of Theorem B.21. $\qquad\square$

## B.4 Proof of Proposition 4.2

Recall that the movement of (2) stays in $\Gamma_\#$, i.e. $|X_t| \in [R_-, R_+]$ and can be characterized as the new model (6), (5).

After applying a time change

$$T = (\eta\lambda)^{\frac{1}{2}} t$$

and writing

$$\bar{\gamma}_t = \eta\lambda\gamma_t = \lambda\eta^{-1}|X_t|^4,$$

we get the following system of equations on $(\overline{X}_T, \bar{\gamma}_T) \in \mathbb{S}^{d-1} \times \mathbb{R}_+$:

$$\mathrm{d}\overline{X}_T = -\bar{\gamma}_T^{-\frac{1}{2}} \overline{\nabla} L(\overline{X}_T) \mathrm{d}T + (\eta\lambda)^{\frac{1}{4}} \bar{\gamma}_T^{-\frac{1}{2}} \bar{\sigma}(\overline{X}_T) \mathrm{d}B_T^d; \tag{44}$$

$$\mathrm{d}\bar{\gamma}_T = (\eta\lambda)^{\frac{1}{2}} \cdot (-4\bar{\gamma}_T + 2\operatorname{Tr}\Sigma(\overline{X}_T)) \mathrm{d}T. \tag{45}$$

To deduce (44), we used the standard fact that for any $a > 0$, $a^{\frac{1}{2}} B_{a^{-1}T}^d$ and $B_T^d$ are equivalent as Wiener processes.

Note $\bar{\gamma}_T$ stays in the fixed interval $[\bar{\gamma}_-, \bar{\gamma}_+] = [\lambda\eta^{-1} R_-^4, \lambda\eta^{-1} R_+^4]$ as long as $X_t \in \Gamma_\#$.

As there are only finitely many basins, we may fix an index $i$ without impact Proposition B.28. In addition, without loss of generality let us assume $L = 0$ on $\Gamma^i$. For our purpose, it might be better to measure the distance to $\Gamma^i$ on $U_{\#,p_0}^i$ where $p_0 > p_1$ is sufficiently small and in particular $U_{\#,p_0}^j$ are disjoint for distinct $j$'s. For $q \geq 0$, we will write $V_{r,q}^i = \{x \in U_{r,p_0}^i, L(x) < q\}$ and $V_{\#,q}^i = \{x \in U_{\#,p_0}^i, L(x) < q\}$.

By Assumption 1.2.(ii), one may manipulate $p_0, q_0, p_1, q_1$ ($p_0 > p_1$, $q_0 > q_1$) such that

$$U_{1,p_1}^i \subseteq V_{1,q_1}^i \subseteq V_{1,q_0}^i \subseteq U_{1,p_0}^i,$$

and thus reformulate Proposition 4.2 as

**Proposition B.28.** *There exists $c > 0$ such that if $q_0 > q_1 > 0$ are fixed, but $q_1$ is sufficiently small compared to $q_0$, then in the regime $\eta \leq O(\lambda) \leq O(1)$,*

$$\lim_{\eta\lambda \to 0} \sup_{(y_0, z_0) \in V_{1,q_1}^i \times [\bar{\gamma}_-, \bar{\gamma}_+]} \mathbb{P}_{(\overline{X}_0, \bar{\gamma}_0) = (y_0, z_0)} \left( \overline{X}_T \text{ remains in } V_{1,q_0}^i \text{ for all } t \in [0, e^{c(\eta\lambda)^{-\frac{1}{2}}}] \right) = 1.$$

*The convergence is uniform with respect to $(y_0, z_0)$.*

*Proof.* In order to apply Theorem B.21 with domain $V_{1,q_0}^i$ and control domain $D = [\bar{\gamma}_-, \bar{\gamma}_+]$, we first make the following remark.

Theorem B.21 is in the setting where the domain is an open neighborhood in $\mathbb{R}^n$, while our current $V_{q_0}^i$ is a neighborhood in the sphere. This is not a problem because unless $L$ is a constant function, $\Gamma_1^i$ is a proper subset of the sphere $\mathbb{S}^{d-1}$. And $V_q^i$ is also a proper subset in $\mathbb{S}^{d-1}$ for small $q$. One can then change coordinates and identify $V_q^i$ with a subset of the Euclidean space.

This converts the problem to the equations (16), (17) with the following dictionary: $\epsilon = (\eta\lambda)^{\frac{1}{2}}$; $\overline{X}_T$ and $\bar{\gamma}_T$ play the roles of $Y_t^\epsilon$, and $Z_t^\epsilon$ respectively; $b(y, z) = -z^{-\frac{1}{2}} \overline{\nabla}(y)$; $\sigma(y, z) = z^{-\frac{1}{2}} \bar{\sigma}(y)$; and $h(\epsilon, y, z) = \epsilon(-4z + 2\operatorname{Tr}\Sigma(y))$.

It remains to check Assumptions B.8 and B.19. We start with the latter. The strictly positivity of $L$ directly follows from the construction of $L$ and the neighborhood $V_{q_0}^i$. The property (1) in Assumption B.19 holds because $b$ is negatively proportional to the gradient $\overline{\nabla}$ of $L$ with respect to the spherical coordinates.

Unfortunately Assumption B.8 doesn't automatically hold as trajectories may escape from $V_{1,q_0}^i$. In order to adapt to this case, we smoothly modify the value of $\sigma$ on $\overline{V_{1,q_0}^i}$ so that it remains unchanged on $\overline{V_{1,q_1}^i}$ and vanishes near $\partial V_{1,q_0}^i$. Then near the boundary, the SDEs (16), (17) become deterministic. Because of Assumption B.19.(1), the trajectories do not escape from $\partial V_{1,q_0}^i$. Moreover, we know that $\bar\gamma_t$ remains in $D = [\bar\gamma_-, \gamma_+]$. This verifies Assumption B.8 for the modified model.

We conclude by applying Theorem B.21 that, for some fixed $\mathcal{I} > 0$, for all initial positions in $V_{1,q_1}^i \times [\bar\gamma_-, \bar\gamma_+]$, the probability that a trajectory $(\overline{X}_T, \bar\gamma_T)$ (with respect to the modified model) leaves $V_{1,q_0}^i \times [\bar\gamma_-, \bar\gamma_+]$ before $T = e^{\frac{\mathcal{I}}{\eta\lambda}} = e^{O((\eta\lambda)^{-\frac{1}{2}})}$ converges to 0 as $\eta\lambda \to 0$. The convergence is in addition uniform with respect to the initial position.

Since such modifications only take place outside $V_{1,q_1}^i \times [\bar\gamma_-, \bar\gamma_+]$, the same statement also holds for the original model. As $V_{1,q_1}^i \subset V_{0,q_0}^i$, we obtain the statement of Proposition B.28 after reparamatrization of variables. $\square$

## C  Unique Katzenberger limit inside each basin

The results from §B.2, stated in the form of Proposition 4.2, guarantee that, after discarding an exponentially small subset of random incidences, the trajectories of (2) stays inside the basin that contains the initial position for exponentially long time $O(e^{C(\eta\lambda)^{-1}})$. We now justify Proposition 4.3

We restart (2) from an initial point, still written as $x_0$ by abuse of notation, in some $U_{\#,p_1}^i$. We now apply a uniform approximation theorem, which is a stronger version of (Li et al., 2022a, Theorem 4.6). We can prove this uniform approximation result because we can strengthen(Katzenberger, 1991, Theorem 6.3) to be a compactness theorem, uniformly with respect to the initial points of the SDE. To describe this result, let $(\Omega^n, \mathcal{F}^n, \{\mathcal{F}_t^n\}_{t\geq0}, \mathbb{P})$ be a filtered probability space, $Z_n$ an $\mathbb{R}^e$-valued cadlag $\{\mathcal{F}_t^n\}$-semimartingale with $Z_n(0) = 0$ and $A_n$ a real-valued cadlag $\{\mathcal{F}_t^n\}$-adapted nondecreasing process with $A_n(0) = 0$. Let $\sigma_n : U \to \mathbb{M}(d, e)$ be continuous with $\sigma_n \to \sigma$ uniformly on compact subsets of $U$. Let $X_n$ be an $\mathbb{R}^d$-valued cadlag $\{\mathcal{F}_t^n\}$-semimartingale satisfying, for all compact $K \in U$,

$$X_n(t) = X_n(0) + \int_0^t \sigma(X_n)dZ_n + \int_0^t -\nabla L(X_n)dA_n \tag{46}$$

for all $t \leq \lambda_n(K)$ where $\lambda_n(K) = \inf\{t \geq 0 | X_n(t-) \notin \mathring{K}$ or $X_n(t) \notin \mathring{K}$ is the stopping time of $X_n$ leaving $K$.

**Theorem C.1.** *Suppose $X_n(0) \in U$, Assumption 3.1, 3.2 and Condition B.2, B.3, B.4, B.5 from (Li et al., 2022a) hold. For any compact $K \subset U$ define $\mu_n(K) = \inf\{t \geq 0 | Y_n(t-) \notin \mathring{K}$ or $Y_n(t) \notin \mathring{K}\}$, then the sequence $\{(Y_n^{\mu_n(K)}, Z_n^{\mu_n(K)}, \mu_n(K))\}$ is relatively compact in $\mathcal{D}_{\mathbb{R}^{d \times e}}[0, \infty) \times [0, \infty)$. If $(Y, Z, \mu)$ is a limit point of this sequence under the Skorohod metric, then $(Y, Z)$ is a continuous semimartingale, $Y(t) \in \Gamma$ for every $t \geq 0$ a.s., $\mu \geq \inf\{t \geq 0 | Y(t) \notin \mathring{K}\}$ a.s. and $Y(t)$ admits*

$$Y(t) = Y(0) + \int_0^{t \wedge \mu} \partial\Phi(Y(s))\sigma(Y(s))dZ(s)$$
$$+ \frac{1}{2} \sum_{i,j=1}^d \sum_{k,l=1}^e \int_0^{t \wedge \mu} \partial_{ij}\Phi(Y(s))\sigma(Y(s))_{ik}\sigma(Y(s))_{jl}d[Z_k, Z_l](s). \tag{47}$$

We will present Assumption B.3, B.4 and Condition B.5, B.6, B.7 and B.8 from (Li et al., 2022a) in Appendix B.

The main difference of Theorem C.1 to (Katzenberger, 1991, Theorem 6.3) is that we allow the initial point $X_n(0)$ to vary within $U$.

**Theorem C.2.** *Let the manifold $\Gamma$ and its open neighborhood $U$ satisfy Assumption 3.1 and 3.2. Let $K \subset U$ be any compact set and $x_{n,0} \in K$ be a sequence of initial points. Consider the SGD formulated in (46) where $X_{\eta_n}(0) \equiv x_{n,0}$. Define*

$$Y_{\eta_n}(t) = X_{\eta_n}(t) - \Phi(X_{\eta_n}(0), A_{\eta_n}(t)) + \Phi(X_{\eta_n}(0))$$

*and $\mu_{\eta_n}(K) = \min\{t \in \mathbb{N} | Y_{\eta_n}(t) \notin \mathring{K}\}$. Then the sequence $\{Y_{\eta_n}^{\mu_n(K)}, Z_{\eta_n}^{\mu_n(K)}, \mu_{\eta_n}(K))\}_{n \geq 1}$ is relatively compact in $\mathcal{D}_{\mathbb{R}^{d \times e}}[0, \infty) \times [0, \infty)$. If $(Y, Z, \mu)$ is a limit point of this sequence, it holds that $Y(t) \in \Gamma$ a.s. for all $t \geq 0$, $\mu \geq \inf\{t \geq 0 | Y(t) \notin \mathring{K}\}$ and $Y(t)$ admits*

$$Y(t) = \int_0^{t \wedge \mu} \partial\Phi(Y(s))\sigma(Y(s))dZ(s) + \frac{1}{2}\int_0^{t \wedge \mu}\sum_{i,j=1}^{d}\partial_{ij}\Phi(Y(s))(\sigma(Y(s))\sigma(Y(s))^\top)_{ij}ds \quad (48)$$

*where $\{W(s)\}_{s \geq 0}$ is the standard Brownian motion.*

*Proof.* The proof of Theorem C.2 follows how (Li et al., 2022a, Theorem B.8) was proved by using (Li et al., 2022a, Lemma B.6) and the standard Katzenberger's theorem (Katzenberger, 1991, Theorem 6.3). One difference is that here not all trajectories stays inside one basin. However, we claim that the probability that trajectories escape the basin goes to zero when $\eta\lambda$ tends to zero. Once this claim is proved, Theorem C.2 is an immediate consequence of (Li et al., 2022a, Lemma B.6) and Theorem C.1.

To prove the claim, we adopt the same idea as in the proof of Theorem 4.2 in Appendix B.4. Although trajectories may escape from level set $V_{1,q_0}^i$, we can smoothly modify the value of $\sigma$ on the closure $\overline{V_{1,q_1}^i}$ so that it remains unchanged on $\overline{V_{1,q_0}^i}$ and vanishes near the boundary $\partial V_{1,q_0}^i$. The SDEs become deterministic, and thus the trajectories of the modified model do not escape from the boundary $\partial V_{1,q_0}^i$. Now, as $\eta\lambda$ tends to zero, by Theorem B.21 the probability of a trajectory of the modified model leaving $\partial V_{1,q_1}^i \times [\bar{\gamma}_-, \bar{\gamma}_+]$ before a fixed $T$ tends to 0. Since such modifications only take place outside $\partial V_{1,q_1}^i \times [\bar{\gamma}_-, \bar{\gamma}_+]$, the same statement holds for the original model. This finishes the proof of the claim. $\qquad\square$

*Proof of Proposition 4.3.* The above uniform version of the Katzenberger's theorem guarantees that, starting from different initial points in the same compact neighborhood of the basin, the distribution of trajectories associated with (2) is still close to that of the Katzenberger's SDE (47). By Proposition 3.1, the latter is mixing towards a unique equilibrium $\nu^i$. Note that even though in Chapter 3, we have only proved it for one basin case, Theorem 4.1 shows that with a large probability, the trajectories do not escape from the basin. For those trajectories, they satisfy a modified SDE equation like before, so that all trajectories do not escape from this basin. At this moment, we can directly apply Proposition 3.1. It follows that within any polynomial time window under consideration, the distribution of trajectories associated with (2) are also mixing towards $\nu^i$. This proves Proposition 4.3. $\qquad\square$

**Assumption C.3.** *(Li et al., 2022a, Assumption 3.1) Assume that the loss $L : \mathbb{R}^d \to \mathbb{R}$ is a $\mathcal{C}^3$ function, and that $\Gamma$ is a $(d - M)$-dimensional $\mathcal{C}^2$-submanifold of $\mathbb{R}^d$ for some integer $0 \leq M \leq d$, where for all $x \in \Gamma$, $x$ is a local minimizer of $L$ and $rank(\nabla^2 L(x)) = M$.*

**Assumption C.4.** *(Li et al., 2022a, Assumption 3.2) Assume that $U$ is an open neighborhood of $\Gamma$ satisfying that gradient flow starting in $U$ converges to some point in $\gamma$, i.e., $\forall x \in U$, $\Phi(x) \in \Gamma$. (Then $\Phi$ is $\mathcal{C}^2$ on $U$ by (Falconer, 1983).)*

**Condition C.5.** *(Li et al., 2022a, Lemma B.2) The integrator sequence $\{A_n\}_{n \geq 1}$ is asymptotically continuous: $\sup_{t > 0}|A_n(t) - A_n(t-)| \Rightarrow 0$ where $A_n(t-) = \lim_{s \to t-} A_n(s)$ is the left limit of $A_n$ at $t$.*

**Condition C.6.** *(Li et al., 2022a, Lemma B.3) The integrator sequence $\{A_n\}_{n \geq 1}$ increases infinitely fast: $\forall \epsilon > 0$, $\inf_{t \geq 0}(A_n(t + \epsilon)) - A_n(t)) \Rightarrow \infty$.*

**Condition C.7.** *((Katzenberger, 1991, Equation 5.1), (Li et al., 2022a, Lemma B.4)) For every $T > 0$, as $n \to \infty$, it holds that*

$$\sup_{0 < t \leq T \wedge \lambda_n(K)} \|\Delta Z_n(t)\|_2 \Rightarrow 0.$$

**Condition C.8.** *((Katzenberger, 1991, Condition 4.2), (Li et al., 2022a, Lemma B.5)) For each $n \geq 1$, let $Y_n$ be a $\{\mathcal{F}_t^n\}$-semimartingale with sample paths in $\mathcal{D}_{\mathbb{R}^d}[0, \infty)$. Assume that for some $\delta > 0$ allowing $\delta = \infty$ and every $n \geq 1$ there exist stopping times $\{\tau_n^m | m \geq 1\}$ and a decomposition of $Y_n - J_\delta(Y_n)$ into a local martingale $M_n$ plus a finite variation process $F_n$ such that $\mathbb{P}[\tau_n^m \leq m] \leq 1/m$, $\{[M_n](t \wedge \tau_n^m) + T_{t \wedge \tau_n^m}(F_n)\}_{n \geq 1}$ is uniformly integrable for every $t \geq 0$ and $m \geq 1$, and*

$$\lim_{\gamma \to 0} \limsup_{n \to \infty} \mathbb{P}\Big[ \sup_{0 \leq t \leq T} (T_{t+\gamma}(F_n) - T_t(F_n)) > \epsilon \Big] = 0,$$

*for every $\epsilon > 0$ and $T > 0$, where $T_t(\cdot)$ denotes total variation on the interval $[0, t]$.*

It was shown in (Li et al., 2022a, Lemma B.6) that for SGD formulated in (46), the sequences $\{A_n\}_{n \geq 1}$ and $\{Z_n\}_{n \geq 1}$ satisfy Condition C.5, C.6, C.7, and C.8. And the landscape of $L$ satisfies Assumption C.3 and C.4. Thus the Katzenberger theorem holds in our case.

