# OpenReview forum: "Fast Equilibrium of SGD in Generic Situations"
_ICLR.cc/2024/Conference — ICLR 2024 poster_

### Official Review · Reviewer_pBQz · 2023-10-28

**Soundness:** 3 good
**Presentation:** 3 good
**Contribution:** 3 good
**Rating:** 6
**Confidence:** 1

**Summary:**

To understand the behaviors of normalization in deep learning, Li et al. (2020) proposes the Fast Equilibrium conjecture: the scale-invariant normalized network, when trained by SGD with $\eta$ learning rate and $\lambda$ weight decay, mixes to an equilibrium in $\tilde{O}(\frac{1}{\eta \lambda})$ steps, as opposed to classical $e^{O((\eta \lambda)^{-1})}$ mixing time. Recent works by Wang & Wang (2022) and Li et al. (2022c) further proved this conjecture under different sets of assumptions.

This paper instead proves the fast equilibrium conjecture in full generality by removing the non-generic assumptions of Wang & Wang (2022) and Li et al. (2022c) that the minima are isolated, that the region near minima forms a unique basin, and that the set of minima is an analytic set. Their main technical contribution is to show that with probability close to 1, in exponential time trajectories will not escape the attracting basin containing its initial position.

**Strengths:**

**Originality:** 1) They first analyze the generality of assumptions used in existing work and then successfully remove non-generic assumptions, which is very important to reduce the gap between the theory and the experiments 2) They use Arnold-Kliemann's condition instead of Kliemann's condition to remove the analyticity assumption 3) They make use of large deviation principle of Dembo-Zeitouni in Dembo & Zeitouni (2010, Chapter 5) instead of the Freidlin-Wentzell’s original theory to show that with very high probability, the trajectory will not escape from the basin in exponential time to remove the unique basin assumption

**Quality:** I'm not familiar with the theoretical techniques but I feel it has good quality.

**Clarity:** It looks quite clear in most cases but needs some modifications for some tiny writing errors.

**Significance:** Okay but not very significant for the following two reasons: 1) It relaxes the assumptions for an existing conjecture instead of discovering some new phenomena, 2) The fast equilibrium conjecture is not as important as many other things in deep learning, such as the generalization ability of deep neural networks, the puzzles in large language models, the interplay among the model, the algorithm, and the data, etc.

**Weaknesses:**

1) They only conduct experiments on MNIST which is almost a linearly-separable dataset, which is not good for deep learning analysis. I suggest the authors conduct the experiments on other more difficult datasets, such as CIFAR-10.
2) There are some tiny errors in the paper, so I'd suggest the authors to proofread their paper more carefully.
- They sometimes use an uncommon citation format for references in some places. For example,
"The works by Bovier et al. and Shi et al. (Bovier et al., 2004; Shi et al., 2020)" may be changed as "Bovier et al. (2004) and Shi et al. (2020)"; "Li et al. made certain assumptions in (Li et al., 2022c)" may be changed as "Li et al. (2022c) made certain assumptions".
- Similarly, they seem use wrong references in some places. For example,
"We now stop assuming Assumption 1.3.(ii) and decompose" => I feel it's Assumption 1.3. (i) instead of Assumption 1.3 (i) since they are talking about removing the unique basin assumption (Assumption 1.3 (i));
"Recall that (Li et al., 2022c)) also assumes Assumption 1.3.(i), but that can be dropped by the discussion in Chapter 3 above." => I feel it's Assumption 1.3 (ii) instead of Assumption 1.3 (i) since Chapter 3 discusses removing the analyticity assumption (Assumption 1.3. (ii));
"Figure 5 shows that V11 and V22 stabilizes near similar but different values," => I think they mean Figure 1 instead of Figure 5 here.

**Questions:**

I have a question about the classical mixing time: You use $e^{O(\eta^{-1})}$ in the abstract but use $e^{O((\eta \lambda)^{-1})}$ in the Introduction. I feel $e^{O((\eta \lambda)^{-1})}$ looks more natural to me. Could you clarify this?

Similarly, $\tilde{O}(1/\eta\lambda)$ in the abstract may need to be changed as $\tilde{O}(\frac{1}{\eta \lambda})$.

**Details Of Ethics Concerns:**

No ethics concerns.

---

> ### Author Response · Authors · 2023-11-15
>
> We would like to thank you for your valuable comments. Some explanations in response to your remarks are below and we will address these concerns in a revised version accordingly.
>
> 1. We are running an experiment with CIFAR 10 and will add it to the revised version.
> 2. We will correct the typos and inaccuracies you pointed out.
> 3. You are right about the exponents in the abstract. They should be $e^{O( (\eta\lambda )^{-1}) } $ and $\tilde{O}(\frac{1}{\eta\lambda})$ respectively. We will fix them.

---

### Official Review · Reviewer_LagX · 2023-11-01

**Soundness:** 4 excellent
**Presentation:** 4 excellent
**Contribution:** 2 fair
**Rating:** 8
**Confidence:** 3

**Summary:**

In this technical paper, the authors deal with fast convergence of networks with normalized steps to the solution.  The authors successfully prove their result and many insights are given on practical training of these networks in the experiments section.

**Strengths:**

This is a very technical paper, very mathematical, and centered around proving one conjecture in the literature. I like this, and I also enjoyed reading the paper: it is very well written and has pleasant notation. What I especially liked is the clear statement of the assumption and remarks 1.4 and 1.5. The authors also showed they mastered the subject with many useful citations and discussions around existing results. Though I did not unfortunately find the time to go through the proofs, the narrative and the impeccable discussion in the main paper leave no doubts on correctness. On a technical level, I was always curious about projections of the Brownian motion on the sphere in the context of Langevin dynamics, so I will for sure get back to this paper in the future to find more details on this. I also have a question, which you can find in the proper section below.

I placed the contribution as fair since I think the result does not explain the fast convergence of normalized networks compared to not-normalized (motivation in the abstract). Please comment on this if you think your result motivates this, I am happy to revise!

**Weaknesses:**

I guess one obvious question is "can you have more experiments". I think this might be silly in this paper, but maybe there is something you can do to reach people outside the very technical domain. One thing I think would be useful is to illustrate the rates known in the literature and the conjecture - maybe with evidence from some datasets and some networks. I think you can attract the interest of many people if you have a headline figure showing the speed of convergence to the stationary distribution and exactly the rate you prove.
A thing I found a bit confusing is going back-and-forward on between $O(1/\eta)$ and $e^{O(\eta^{-1})}$ results. That got me a bit thinking and I have a question (below).

**Questions:**

1) This is something I am pretty sure I could solve on my own with a bit of thinking, but I guess it hints to lack of clarity in some parts of the intro: I find a bit of contradiction between the sentences (1, abstract) "scale-invariant normalized network, mixes to an equilibrium in
$O(1/\eta\lambda)$ steps, as opposed to classical $e^{O(\eta^{−1})}$ mixing time" and (2, intro) "When normalization is used,  the effective learning rate for the renormalized parameter vector will stabilize around $O((\eta\lambda^{-1/2} )$ and in consequence $e^{O((\eta\lambda)^{-1}}$ is replaced with $e^{O((\eta\lambda)^{-1/2}}$". I think this is a bit unclear, can you please explain?

2) From my SDE knowledge, I always understood that convergence to the stationary distribution is dominated by the drift. What is the convergence rate -- to a local minimum -- if you drop the noise term? I think providing an analysis of this setting will certainly help the reader understand the proof in a simplified setting.

3) The SDE you study certainly is a model for SGD in the setting you study. However, I am a bit worried about the noise structure. Is there some guarantee in previous literature that constant gradient noise in the not-normalized setting translates to the noise projection structure you study in the normalized case? I am talking about formula 4-5 in comparison to the discrete update.

---

> ### Author Response · Authors · 2023-11-15
>
> We would like to thank you for your valuable comments. Some explanations in response to your remarks are below and we will address these concerns in a revised version accordingly.
>
> 0. Response to [“ I think you can attract the interest of many people if you have a headline figure showing the speed of convergence to the stationary”]. Thanks for this suggestion. Experiments where the empirically observed rates of convergence are polynomial were contained in the original paper (Li et. al 2020) where the Fast Equilibrium Conjecture was first asked. We will add references to figures from that paper.
> 1. Response to Question 1. Thanks for pointing out the unclarity of the writing here. The difference between the polynomial and exponential times is explained in the response 2 below. The difference between $(\eta\lambda)^{-1}$ and $(\eta\lambda)^{-\frac12}$, is due to the fact that (Bovier et al., 2004; Shi et al., 2020) studied neural networks without normalization, which involved an analysis on $R^d$. With normalization and weight decay together, (Li et al. 2020) proved that the problem, via the polar coordinate decomposition written as (4) and (5) in our paper, can be considered as a problem on $S^{d-1}$. After this polar decomposition step, the effective learning rate is $O((\eta\lambda)^{\frac12})$, which results in replacements of exponents appearing in convergence times.
> 2. Response to Question 2. In this case, the observed convergence to equilibrium is dominated by the diffusion. The drift regulates how fast the trajectories fall near the bottom of a basin (the “descent stage”), i.e. how fast the loss function value converges. However, the trajectories then start a random walk all around the bottom of the basin. This is the “diffusion stage”, whose time scale is regulated by the diffusion term. (The third stage takes an exponentially long time when trajectories leave the basin to nearby basins but this cannot be observed in practice because of the exponential time scale.)
> 3. Response to Question 3.  Even though some earlier papers assumed a constant gradient noise structure, our current paper does not need to assume constant gradient noise. In fact, equations (4) and (5) were proved in (Li et al. 2022) and didn’t require constant noise structure.

---

### Official Review · Reviewer_yR6a · 2023-11-02

**Soundness:** 3 good
**Presentation:** 4 excellent
**Contribution:** 3 good
**Rating:** 6
**Confidence:** 3

**Summary:**

This paper provides a strengthened proof of the fast equilibrium conjecture that was proved in the previous works (Wang &Wang (2022); Li et al. (2022c)) by removing the non-generic assumptions of the unique basin and that the set of minima is an analytic set. In order to remove these additional assumptions, this paper mainly adopts a purely probabilistic method rather than the spectral analysis that was used in the previous works.

Toward this goal, this work shows that trajectories would not escape from the initial basin in exponential time.

**Strengths:**

1. This paper is clearly organized and well-structured so that it is easy for the readers to grasp the main contributions of this work.
2. I like the fact that this work provides solid and well-supported arguments (Remarks 1.4 and 1.5) that Assumptions 1.2 are natural and Assumptions 1.3 are non-generic. These motivate removing these less natural assumptions well and signifies the contribution of this work.
3. The main result that the Fast Equilibrium conjecture holds without the assumptions of the unique basin and that the set of minima is an analytic set is significant, and it contributes well to the theoretical understanding of the effects of normalization layers.
4. While I did not check all the proof details, I followed the proof at a high level.

**Weaknesses:**

It could be good to add some theoretical reasonings (in addition to being natural) about why Assumption 1.2 might be essential to prove the Fast Equilibrium conjecture.

**Questions:**

1. Will the noise structure affect the convergence rate?
2. Would it be possible to achieve a similar result if $L$ had a homogeneity degree  $> 0$?
3. Are all the remaining three assumptions essential to prove the Fast Equilibrium conjecture? Would it be possible to even weaken the assumptions?
4. Minor:
In the first line in Section 4, "Assumption 1.3. (ii)" -> "Assumption 1.3. (i)"?

---

> ### Author Response · Authors · 2023-11-15
>
> We would like to thank you for your valuable comments. Some explanations in response to your remarks are below and we will address these concerns in a revised version accordingly.
>
> 1. Response to [Question 1]. The noise structure does affect the convergence rate. But as long as Assumption 1.2 is satisfied, the noise structure wouldn’t affect the asymptotic order of the convergence rate.
> 2. Response to [Question 2]. This is a very interesting question worth further exploration, but is not covered by our result as we rely on a polar decomposition of SDEs (written as (4), (5) and (6) in the original version, the one before current revision), from (Li et al. 2020)  that depends on the current homogeneity structure.
> 3. Response to [Weaknesses] and [Question 3]. You are absolutely right that the conditions in Assumption in 1.2 are essential for our analysis and we will add explanations in a revision. To be precise:
>  * Assumption 1.2 (i) is essential because without this assumption, the SDE would not be equivalent to a SDE on the sphere $S^{d-1}$ (written as (6) in the original version, the one before current revision), which is crucial to our analysis. Without this assumption, similar analysis can probably be formulated on $R^d$ instead of the $S^{d-1}$ coordinate but there will be new technical obstacles to overcome. Since the original fast equilibrium conjecture was asked for normalized neural nets, we restrict our study to the current setting.
>  * Assumption 1.2 (ii) is important because if not, a trajectory may stay near a critical point (for example a saddle point) for a very long period of time, and thus it would not be able to converge within a polynomial time.
> * The reason why we need Assumption 1.2 (iii) is that if the span is not the whole tangent space, but instead a subspace of the tangent space, then the diffusion will be restrained to this subspace, which a priori may be very fractal and existing mathematical theory is not sufficient to guarantee a unique equilibrium in limit.
> 4. We will fix this typo.

---

> > ### Comment · Reviewer_yR6a · 2023-11-22
> >
> > Thanks for the responses. My questions are addressed, and I keep my rating for this work.

---

### Official Review · Reviewer_jkii · 2023-11-10

**Soundness:** 3 good
**Presentation:** 2 fair
**Contribution:** 3 good
**Rating:** 6
**Confidence:** 3

**Summary:**

This paper proves the fast equilibrium conjecture for SGD on neural nets with normalization layers in a more general setting than previous works Li et al., 2022c and Wang & Wang, 2022. Specifically, it shows that the conjecture still holds without the unique basin and analyticity assumptions made in Li et al., 2022c. The theoretical results are further supported by experiments.

**Strengths:**

This paper is mathematically solid. It extends the conditions for the fast equilibrium conjecture to a more general setting, making a good technical contribution to the community.

**Weaknesses:**

1. Referring to the second experiment as "stochastic weight averaging" (SWA) is inappropriate, as SWA averages the model parameters at different iterations along the same trajectory. Conversely, the approach in this paper averages the parameters at the same iterations from different trajectories.

2. Missing discussion of related works, which

   - argue that the iterates stay in the same basin for a significant amount of time when starting from the same initialization.
     - Frankle, J., Dziugaite, G. K., Roy, D., & Carbin, M. (2020, November). Linear mode connectivity and the lottery ticket hypothesis. In *International Conference on Machine Learning* (pp. 3259-3269). PMLR.
     - Gupta, V., Serrano, S. A., & DeCoste, D. (2019, September). Stochastic Weight Averaging in Parallel: Large-Batch Training That Generalizes Well. In *International Conference on Learning Representations*.

   - Also analyze the dynamics of SGD / Local SGD near the manifold of minimizers
     - Damian, A., Ma, T., & Lee, J. D. (2021). Label noise sgd provably prefers flat global minimizers. *Advances in Neural Information Processing Systems*, *34*, 27449-27461.
     - Gu, X., Lyu, K., Huang, L., & Arora, S. (2022, September). Why (and When) does Local SGD Generalize Better than SGD?. In *The Eleventh International Conference on Learning Representations*.

3. The paper is abstract in its current form. It would be beneficial if the authors could provide specific examples where the removed assumptions may be too restrictive.

BTW, I did not have time to check the proof. So, my final rating will be influenced by the evaluations of the other reviewers.

**Questions:**

N/A

---

> ### Author Response · Authors · 2023-11-15
>
> We would like to thank you for your valuable comments. Some explanations in response to your remarks are below and we will address these concerns in the revised version accordingly.
> 1. We would change the name of SWA to “weight average over trajectories”.
> 2. We will add the references you pointed out.
> 3. The previous assumption of analyticity is restrictive because the regularity of the loss function is decided by that of the activation function. Even though popular activation functions such as Sigmoid are analytic, a priori one could use smooth but not analytic functions. The one basin assumption is restrictive as we do not see empirical evidence of proof that $L$ only has one basin. In fact, the experiments at the end of the paper suggests that there are multiple basins.

---

> > ### Comment · Reviewer_jkii · 2023-11-16
> > **Response acknowledged**
> >
> > The authors' response has well addressed my concerns. I will keep my positive rating.

---

### Author Response · Authors · 2023-11-15

We thank all referees for their careful reading and helpful remarks. We will make the changes suggested by the referees in the revised version. In particular, we will:
* avoid the inaccurate naming of SWA;
* expand discussions about essentialness and restrictiveness of both removed and remaining assumptions;
* add references to experiments from previous literature that supports the validity of fast equilibrium conjecture, and extend our experiments to include CIFAR-10 dataset;
* clarify the impact of the diffusion noise on convergence times.

---

### Meta-Review · Area_Chair_9mFq · 2023-12-05

**Metareview:**

The paper contributes to the literature on fast mixing of SGD when applied to the training of deep neural nets with layer normalization. This line of work was started with a conjecture posed  in Li et al. (2020), which posits that layer normalization leads to polynomial $\tilde{O}(1/(\eta\lambda))$ (as opposed to the exponential $\tilde{O}(\exp(1/(\eta \lambda)))$) mixing time, where $\eta$ is the learning rate and $\lambda$ the weight decay parameter. The conjecture was proved in recent works (Wang & Wang (2022) and Li et al. (2022)), but under fairly strong assumptions. This work proves the same conjecture under a much less stringent set of assumptions, and with different techniques. More specifically, the paper removes the assumptions that the minima are isolated, that the region near minima forms a unique basin, and that the set of minima is an analytic set. The paper provides solid theoretical contributions to this area and will be of value to the broad ML community.

The reviewers noted a few minor presentation issues and suggested adding more details about the assumptions and adding more experiments. The paper would certainly benefit from taking these into account.

**Justification For Why Not Higher Score:**

It is a nice paper, but I am not sure it is appropriate for spotlight, as it builds on the existing line of work. I reserve spotlight/oral for papers that are particularly insightful, conceptually novel, or that lead to much higher technical advances.

**Justification For Why Not Lower Score:**

I think the work is a valuable contribution to the community that many people would be interested in, and it is sufficiently technically nontrivial to be accepted.

---

### Decision · Program_Chairs · 2024-01-16

Accept (poster)